# Complete set of quasi-conserved quantities for spinning particles around Kerr

Geoffrey Compère[*] and Adrien Druart[†]

*Université Libre de Bruxelles, Gravitational Wave Centre,*
*International Solvay Institutes, CP 231, B-1050 Brussels, Belgium*
(Dated: October 18, 2021)

We revisit the conserved quantities of the Mathisson-Papapetrou-Tulczyjew equations describing the motion of spinning particles on a fixed background. Assuming Ricci-flatness and the existence of a Killing-Yano tensor, we demonstrate that besides the two non-trivial quasi-conserved quantities, i.e. conserved at linear order in the spin, found by Rüdiger, non-trivial quasi-conserved quantities are in one-to-one correspondence with non-trivial mixed-symmetry Killing tensors. We prove that no such stationary and axisymmetric mixed-symmetry Killing tensor exists on the Kerr geometry. We discuss the implications for the motion of spinning particles on Kerr spacetime where the quasi-constants of motion are shown not to be in complete involution.

## CONTENTS

## I. INTRODUCTION

The prospective observation of extreme mass ratio inspirals (EMRIs) with the LISA mission strongly motivates the modeling of black hole binaries in the small mass ratio regime [1, 2]. In the self-force formalism, the leading order motion in the small mass ratio expansion reduces to the motion of a spinning particle orbiting the Kerr black hole [3, 4]. The equations of motion of such a system were established by Mathisson [5] and Papapetrou [6], which can be completed with the spin supplementary condition of Tulczyjew [7, 8]. In astrophysical scenarios, the spin scales as the mass ratio times the mass of the Kerr black hole squared as a result of the maximally spinning bound [9, 10]. It implies that quadratic effects in the spin are of the same order of magnitude as second order self-force effects for EMRIs which are pertinent for LISA observations [11]. Moreover, since they appear at 2PN order in the post-Newtonian expansion [12–17] (see [18, 19] for the 3PN order and [20] for a recent status) quadratic effects are generically relevant for gravitational waveform modeling of compact binaries.

The conservation of the mass and spin magnitude being taken into account, the Mathisson-Papapetrou-Tulczyjew (MPT) equations can be written in terms of an Hamiltonian system possessing four degrees of freedom [21]. The energy and angular momentum provide two immediate first integrals of motion. Two additional quantities linear in the spin vector were found by Rüdiger [22, 23]. They are quasi-conserved, *i.e.*, conserved at linear order in the spin or, equivalently, admitting an evolution along the orbit (at least) quadratic in the spin. In fact, without further quadratic corrections, Rüdiger's quasi-constants of motion are not conserved at quadratic order in the spin [24].

The unanswered question of the existence of other independent quasi-conserved quantities for the MPT equations led to an undetermined status of the role of integrability and by opposition, chaos, in the dynamics of spinning particles around Kerr. While chaos has been established to appear at second order in the spin [25], numerical simulations suggest that no chaos occurs at linear order in the spin [25–27]. From the solution of the Hamilton-Jacobi equations at linear order in the spin, one can infer that chaotic motion at linear order is negligible [28]. In this paper, we will relate the existence of new quasi-conserved quantities homogeneously linear in the spin to the existence of a new tensorial structure on the background, that we will refer to as a *mixed-symmetry Killing tensor*. This result will apply to the Kerr background and more generally to Ricci-flat spacetimes admitting a Killing-Yano tensor. We will demonstrate that under the assumption of stationarity and axisymmetry, no such non-trivial structure exists on the Schwarzschild background and no non-trivial mixed-symmetry Killing tensor on Kerr can be con-

* geoffrey.compere@ulb.be
† Adrien.Druart@ulb.be

structed from deformations of trivial mixed-symmetry Killing tensors on Schwarzschild. In addition, we will derive that Liouville integrability does not hold around the Kerr background at linear order in the spin since the quasi-conserved quantities are not in involution.

The rest of the paper is organized as follows. In Section II, we review the MPT equations governing the motion of spinning test-particles in a fixed background. In Section III, we review the procedure formulated by Rüdiger [22, 23] to build constants of motion of the MPT equations. We explicit the set of constraints that must be fulfilled for an invariant at most linear in the spin to exist. Conservation at linear order of such invariant requires to solve only two constraints. We simplify the second, most difficult, constraint in Section IV. We subsequently particularize our setup in Section V to spacetimes admitting a Killing-Yano (KY) tensor. After deriving some general properties of KY tensors, we will prove a cornerstone result for the continuation of our work, which we will refer to as the *central identity*. Building on all previous sections, we will solve the aforementioned constraint for Ricci-flat (vacuum) spacetimes possessing a KY tensor in Section VI. This will enable us to study in full generality the quasi-invariants for the MPT equations that are quadratic in the combination of spin and momentum. On the one hand, we recover Rüdiger's results [22, 23]. On the other hand, we prove that the existence of any further quasi-invariant, which is then necessarily homogeneously linear in the spin, reduces to the existence of a non-trivial mixed-symmetry Killing tensor on the background. The significance of this result is examined for spinning test-particles in Kerr spacetime in the final Section VII. We show that a stationary and axisymmetric non-trivial mixed-symmetry Killing tensor does not exist on the Kerr geometry. Consequently, an additional independent quasi-constant of motion for the linearized MPT equations does not exist. As detailed in Appendix A the linearized MPT integrals of motion are not in involution, which implies that the system is not integrable in the sense of Liouville.

*Conventions and notations.* We place ourselves within the framework of General Relativity. Therefore, we will always consider a 4d Lorentzian manifold equipped with a metric $g_{\mu\nu}$. The metric signature is chosen to be $(-+++)$. Lowercase Greek indices run from 0 to 3 and denote spacetime indices. Lowercase Latin indices represent tetrad indices. The Einstein summation convention is used everywhere. $\nabla_\alpha$ denotes the Levi-Civita connexion, the Riemann tensor is defined such that $[\nabla_\alpha, \nabla_\beta] A_\mu = -R^\lambda{}_{\mu\alpha\beta} A_\lambda$ and the Ricci tensor is defined as $R_{\mu\nu} = R^\lambda{}_{\mu\lambda\nu}$.

## II. MOTION OF SPINNING TEST-PARTICLES IN GENERAL RELATIVITY

Let us consider the motion of a object described by the stress-energy tensor $T_{\mu\nu}$ in a background metric $g_{\mu\nu}$. We

have here in mind the motion of a "small" astrophysical object (stellar mass black hole or neutron star) around a hypermassive black hole. In this situation, the former can be viewed as a perturbation of the spacetime geometry created by the later. Furthermore, if the small object is *compact* (*i.e.* if its typical size is much smaller than the typical lengthscale describing the binary system), its internal structure can be fully described in terms of an infinite collection of multipole moments defined on a worldline $X^\mu(\tau)$ [5, 29, 30]:

$$\int_{x^0=\text{constant}} \mathrm{d}^3 x \sqrt{-g} T^{\mu\nu} \delta x^{\alpha_1} \dots \delta x^{\alpha_n} \tag{1}$$

where $\delta x^\mu \triangleq x^\mu - X^\mu(\tau)$ with $\tau$ being the small object's proper time. This representation is usually called the *gravitational skeletonization*.

Hereafter, we will restrict our setup to the *pole-dipole* approximation, which consists into neglecting all the moments but the two first ones, namely the linear momentum $p^\mu$ and the skew-symmetric[1] spin-dipole $S^{\mu\nu}$:

$$p^\mu \triangleq \int_{x^0=\text{constant}} \mathrm{d}^3 x \sqrt{-g} T^{\mu 0}, \tag{2}$$

$$S^{\mu\nu} \triangleq \int_{x^0=\text{constant}} \mathrm{d}^3 x \sqrt{-g} \left( \delta x^\mu T^{\nu 0} - \delta x^\nu T^{\mu 0} \right). \tag{3}$$

Physically, this corresponds to upgrading the test-particle geodesic motion in order to include the effects due to the small body spin, while still neglecting higher moments (quadrupole and higher) due to its more refined internal structure.

### A. Mathisson-Papapetrou equations

The equations of motion for such a spinning test-particle can be worked out using the conservation of the stress tensor $\nabla_\mu T^{\mu\nu} = 0$. This leads to the so-called *Mathisson-Papapetrou equations of motion* (or MP equations for short) [5, 6]:

$$\frac{\mathrm{D} p^\mu}{\mathrm{d}\lambda} = -\frac{1}{2} R^\mu{}_{\nu\alpha\beta} v^\nu S^{\alpha\beta}, \tag{4}$$

$$\frac{\mathrm{D} S^{\mu\nu}}{\mathrm{d}\lambda} = 2 p^{[\mu} v^{\nu]} \tag{5}$$

where we defined the tangent vector to the worldline as $v^\mu \triangleq \frac{\mathrm{d}X^\mu}{\mathrm{d}\lambda}$ ($\lambda$ being any affine parameter) and where $\frac{\mathrm{D}}{\mathrm{d}\lambda} \triangleq v^\mu \nabla_\mu$ is the covariant derivative along the world-

---

[1] The symmetric part of the dipole moment is vanishing when the worldline is chosen as the body's center of mass, *i.e.* when the spin supplementary condition will be enforced. For more details, see *e.g.* the excellent review [31].

line. We also introduce the notations

$$\mathfrak{m} \triangleq -p^\mu v_\mu, \tag{6}$$

$$\mu^2 \triangleq -p^\mu p_\mu, \tag{7}$$

$$\mathcal{S}^2 \triangleq \frac{1}{2} S^{\mu\nu} S_{\mu\nu}. \tag{8}$$

Here, $\mu^2$ is the dynamical rest mass of the object, *i.e.* the mass of the object measured by an observer in a frame where the spatial components of the linear momentum $p^i$ do vanish; $\mathfrak{m}$ will be referred to as the *kinetic mass* and $\mathcal{S}$ as the *spin parameter*.

At this point, let us emphasize that the linear momentum is no aligned with the velocity, and thus not tangent to the worldline, since

$$p^\mu = \frac{1}{v^2} \left( v_\alpha \frac{DS^{\mu\alpha}}{d\lambda} - \mathfrak{m} v^\mu \right). \tag{9}$$

The dynamical and kinetic masses are in general *not* constants of the motion: in fact, using the MP equations (4)-(5), one can show that

$$\frac{d\mathfrak{m}}{d\lambda} = -\frac{1}{v^2} \frac{Dv_\alpha}{d\lambda} v_\beta \frac{DS^{\alpha\beta}}{d\lambda}, \tag{10}$$

$$\frac{d\mu}{d\lambda} = -\frac{1}{\mu \, \mathfrak{m}} p_\alpha \frac{Dp_\beta}{d\lambda} \frac{DS^{\alpha\beta}}{d\lambda}. \tag{11}$$

Similarly, the evolution equation for the spin parameter reads as

$$\frac{d(\mathcal{S}^2)}{d\lambda} = 2 S_{\mu\nu} p^\mu v^\nu. \tag{12}$$

### B. The spin supplementary condition

The motion of the spinning test-particle is described by the fourteen dynamical quantities $v^\mu$, $p^\mu$ and $S^{\mu\nu} = S^{[\mu\nu]}$. However, we only have in our possession ten differential equations, namely the MP equations (4)–(5). The system is consequently not closed, and we are left with four extra dynamical quantities. These four functions can be identified with the worldline $X^\mu$ along which we are defining the particle's multipole moments. We will choose it as being the worldline describing the position of the body's center-of-mass as seen by an observer of 4-velocity proportional to $p^\mu$. This is practically implemented by enforcing both a choice of proper time and the (covariant) *Tulczyjew spin supplementary conditions* (SSCs) [7, 8]

$$S^{\mu\nu} p_\nu = 0. \tag{13}$$

The SSCs form a set of three additional constraints since a contraction with $p_\mu$ leads to a trivial identity. In what follows, we will choose the affine parameter driving the evolution as the particle's proper time, $\lambda = \tau$. This enforces the 4-velocity to be normalized,

$$v^\mu v_\mu = -1, \tag{14}$$

which thereby guarantees its timelike nature along the evolution of the system. These conditions consequently close our system of equations which then call the MPT equations. These conditions fix uniquely the worldline and allows to inverting Eq. (9) in order to express the 4-velocity as a function of the linear momentum. It can be shown [24, 32, 33] that

$$v^\mu = \frac{\mathfrak{m}}{\mu^2} \left( p^\mu + \frac{D^\mu{}_\alpha p^\alpha}{1 - \frac{d}{2}} \right) \tag{15}$$

where we defined[2]

$$D^\nu{}_\beta \triangleq \frac{1}{2\mu^2} S^{\nu\alpha} R_{\alpha\beta\gamma\delta} S^{\gamma\delta}, \tag{16}$$

$$d \triangleq D^\alpha{}_\alpha. \tag{17}$$

The condition (14) allows to algebraically solve for $\mathfrak{m}$ using Eq. (15):

$$\mathfrak{m}^2 = \mu^2 \left( 1 - \frac{D^\mu{}_\alpha p^\alpha D_{\mu\beta} p^\beta}{\mu^2 (1 - d/2)^2} \right)^{-1}. \tag{18}$$

The positivity of $\mathfrak{m}$ and the condition $d < 2$ are not a consequence of the MPT equations but we will enforce these conditions for physical reasons. It will guarantee that the MPT equations perturbatively correct the geodesic motion with spin couplings[3]. In what follows we will always assume that $\mathfrak{m} > 0$ and $d < 2$.

A direct consequence of the SSC conditions is that the spin parameter is constant,

$$\frac{d(\mathcal{S}^2)}{d\tau} = 0. \tag{19}$$

Finally, by differentiating the SSC conditions and plugging them into the rest mass evolution equation (11), it is easy to show that the latter is also a constant of the motion,

$$\frac{d\mu}{d\tau} = 0. \tag{20}$$

### C. The spin vector

Having imposed the Tulczyjew conditions, all the information contained in the spin-dipole tensor can be re-

---

[2] Our definition of $D^\mu{}_\alpha$ differs by a global '$-$' sign from the one provided in [24] due to the convention chosen for the signature.

[3] For a proposal of a completion of the MPT equations with improved ultra-relativistic behavior, see [34, 35].

cast into a *spin vector* $S^\mu$. Indeed, if one defines[4]

$$S^\alpha \triangleq \frac{1}{2}\varepsilon^{\alpha\beta\gamma\delta}\hat{p}_\beta S_{\gamma\delta} \tag{21}$$

where $\hat{p}^\mu \triangleq \frac{p^\mu}{\mu}$ (yielding $\hat{p}^2 = -1$), one can invert the previous relation in order to rewrite $S^{\alpha\beta}$ in terms of $S^\alpha$:

$$S^{\alpha\beta} = -\varepsilon^{\alpha\beta\gamma\delta}\hat{p}_\gamma S_\delta. \tag{22}$$

This can be easily checked using the identity [36]

$$\varepsilon^{\alpha_1...\alpha_j\alpha_{j+1}...\alpha_n}\varepsilon_{\alpha_1...\alpha_j\beta_{j+1}...\beta_n} = -(n-j)!\,j!\,\delta^{[\alpha_{j+1}...\alpha_n]}_{\beta_{j+1}...\beta_n} \tag{23}$$

which is valid in any Lorentzian manifold. We make use of the shortcut notation $\delta^{\mu_1...\mu_N}_{\nu_1...\nu_N} \triangleq \delta^{\mu_1}_{\nu_1}...\delta^{\mu_N}_{\nu_N}$. By definition, the spin vector is automatically orthogonal to the 4-impulsion:

$$p_\mu S^\mu = 0. \tag{24}$$

Finally, also notice that the spin parameter is simply the squared norm of the spin vector, $\mathcal{S}^2 = S^\alpha S_\alpha$ .

### D. Independent dynamical variables

Let us now summarize the independent dynamical variables of our system. Under the SSCs (13), one can write

$$\mu = \mu(p^\alpha) = \sqrt{-p_\alpha p^\alpha}, \tag{25}$$
$$\mathfrak{m} = \mathfrak{m}(p^\alpha, S^\alpha), \tag{26}$$
$$S^{\mu\nu} = S^{\mu\nu}(p^\alpha, S^\alpha) = -\varepsilon^{\mu\nu\alpha\beta}\hat{p}_\alpha S_\beta, \tag{27}$$
$$v^\mu = v^\mu(p^\alpha, S^\alpha). \tag{28}$$

The explicit expression for $v^\mu$ will be worked out in the following subsection. Consequently, the system can be fully described in terms of the twelve dynamical variables $x^\mu$, $p^\mu$ and $S^\mu$. However, the four components of the spin vector $S^\mu$ are not independent, since they are subjected to the orthogonality condition $p^\mu S_\mu = 0$. At the end of the day, we are left with eleven independent variables. From an Hamiltonian perspective, after imposing the constraint on the Hamiltonian $H = -\mu^2/2$ the system $(x^\mu, p^\mu)$ admits 3 degrees of freedom and taking into account the consistency of the spin $S^2$, the spin vector $S^\mu$ admits one further degree of freedom, leading to 4 degrees of freedom [21], see further discussion in Section VII D.

The fact that all the components of $S^\mu$ are not independent will lead to complications when we will seek to build invariants, as detailed in Section III. To overcome this difficulty, we introduce

$$\Pi^\mu_\nu \triangleq \delta^\mu_\nu + \hat{p}^\mu\hat{p}_\nu, \tag{29}$$

the projector onto the hypersurface orthogonal to $p^\mu$. It can be directly checked from the definition that $\Pi^\mu_\nu$ satisfies the properties:

$$\text{(I)} \quad \Pi^\mu_\nu \Pi^\nu_\rho = \Pi^\mu_\rho, \tag{30}$$
$$\text{(II)} \quad \Pi^\mu_\nu\, p^\nu = 0, \tag{31}$$
$$\text{(III)} \quad \Pi^\mu_\alpha\, S^{\alpha\nu} = S^{\mu\nu}. \tag{32}$$

We introduce the *relaxed spin vector* $s^\alpha$ from

$$S^\alpha \triangleq \Pi^\alpha_\beta s^\beta \tag{33}$$

where the part of **s** aligned with **p** is left arbitrary, but is assumed (without loss of generality) to be of the same order of magnitude. It ensures the relation $\mathcal{O}(s) = \mathcal{O}(\mathcal{S})$ to hold (where $s^2 \triangleq s_\alpha s^\alpha$). While working out the conservation constraints, one will often encounter the spin vector antisymmetrized with **p**, in expressions of the type $p^{[\mu}S^{\nu]}$. In that case, we will write $p^{[\mu}S^{\nu]} = p^{[\mu}s^{\nu]}$ thereby replacing the (constrained) **S** by the (independent) variables **s**.

### E. A convenient expression for the 4-velocity

In the remaining of this section, we will derive a convenient expression of the 4-velocity in terms of the impulsion and the spin vector. Let us first introduce some definitions. Given any tensor **A**, one can define the left and the right Hodge duals as

$$^*A_{\mu\nu\alpha_1...\alpha_p} \triangleq \frac{1}{2}\varepsilon_{\mu\nu}{}^{\rho\sigma}A_{\rho\sigma\alpha_1...\alpha_p}, \tag{34}$$
$$A^*_{\alpha_1...\alpha_p\mu\nu} \triangleq \frac{1}{2}\varepsilon_{\mu\nu}{}^{\rho\sigma}A_{\alpha_1...\alpha_p\rho\sigma}. \tag{35}$$

They correspond, respectively, to the Hodge dualization on the two first, resp. the two last, indices of **A**. The definition of the bidual $^*\mathbf{A}^*$ follows directly from the definitions above. Finally, given the product of two antisymmetrized vectors, one can similarly define

$$l^{[\mu}m^{\nu]*} \triangleq \frac{1}{2}\varepsilon^{\mu\nu\rho\sigma}l_\rho m_\sigma. \tag{36}$$

The dual tensors obey the following properties:

$$\text{(I)} \quad A^*_{\mu\nu} = {}^*A_{\mu\nu}, \tag{37}$$
$$\text{(II)} \quad A^{**}_{\mu\nu} = {}^{**}A_{\mu\nu} = -A_{\mu\nu}, \tag{38}$$
$$\text{(III)} \quad {}^*A_{[\mu\nu]}B^{[\mu\nu]} = A_{[\mu\nu]}{}^*B^{[\mu\nu]}. \tag{39}$$

For any metric we have

$$^*R^\alpha{}_{\mu\alpha\nu} = \frac{1}{2}\varepsilon^\alpha{}_\beta{}^{\beta\gamma}R_{[\beta\gamma\alpha]\nu} = 0. \tag{40}$$

---

[4] The convention chosen here differs from the one of [22, 24] by a global '−' sign.

Bianchi's identities $R_{\alpha\beta[\mu\nu;\sigma]} = 0$ can be equivalently written as

$$R^{*}{}_{\alpha\beta\gamma}{}^{\sigma}{}_{;\sigma} = 0. \tag{41}$$

Let us turn to the more specific context of the MPT theory. Using all the previous definitions, it is not complicated to check that the following properties hold:

$$(\text{I}) \quad S^{\alpha\beta} = 2S^{[\alpha}\hat{p}^{\beta]*}, \tag{42}$$

$$(\text{II}) \quad \Pi^{\alpha[\beta}S^{\gamma]*} = S^{\alpha[\beta}\hat{p}^{\gamma]}. \tag{43}$$

This allows us to write

$$D^{\mu}{}_{\alpha}p^{\alpha} = \frac{1}{2\mu}S^{\mu[\nu}\hat{p}^{\alpha]}R_{\nu\alpha\rho\sigma}S^{\rho\sigma} \tag{44}$$

$$= \frac{1}{\mu}\Pi^{\mu[\nu}S^{\alpha]*}R_{\nu\alpha\rho\sigma}S^{[\rho}\hat{p}^{\sigma]*} \tag{45}$$

$$= \frac{1}{\mu}\Pi^{\mu\alpha*}R^{*}_{\alpha\beta\gamma\delta}S^{\beta}S^{\gamma}\hat{p}^{\delta}. \tag{46}$$

Similarly, one can show that

$$d = -\frac{2}{\mu^2}{}^{*}R^{*}_{\alpha\beta\gamma\delta}S^{\alpha}\hat{p}^{\beta}S^{\gamma}\hat{p}^{\delta}. \tag{47}$$

Putting all the pieces together, one can proceed to the desired rewriting of Eq. (15):

$$\frac{\mu^2}{\mathfrak{m}}\left(1 - \frac{d}{2}\right)v^{\mu} = \left(1 - \frac{d}{2}\right)p^{\mu} + D^{\mu}{}_{\alpha}p^{\alpha} \tag{48}$$

$$= p^{\mu} - \frac{1}{\mu}(\hat{p}^{\mu}\hat{p}^{\alpha} - \Pi^{\mu\alpha}){}^{*}R^{*}_{\alpha\beta\gamma\delta}S^{\beta}S^{\gamma}\hat{p}^{\delta} \tag{49}$$

$$= p^{\mu} + \frac{1}{\mu}g^{\mu\alpha*}R^{*}_{\alpha\beta\gamma\delta}S^{\beta}S^{\gamma}\hat{p}^{\delta} \tag{50}$$

$$= p^{\mu} + \frac{1}{\mu}{}^{*}R^{*\mu}{}_{\beta\gamma\delta}S^{\beta}S^{\gamma}\hat{p}^{\delta}. \tag{51}$$

This relation will be a fundamental building block of the forthcoming computations.

### F. MPT equations at linear order in the spin

We take a break and look back at our fundamental motivation: extreme mass-ratio inspirals involving a spinning secondary. The most important parameter for describing an EMRI is the ratio between the mass $\mu$ of the secondary and the mass $M$ of the primary, which is by assumption a small number:

$$\eta \triangleq \frac{\mu}{M} \ll 1. \tag{52}$$

Let us assume that the EMRI central object is a Kerr black hole. As detailed e.g. in [25], the spin term in the MPT equations for such a background spacetime scales as

$$\frac{S}{\mu M} \leq \frac{\mu^2}{\mu M} = \eta, \tag{53}$$

in any astrophysically realistic situation. Therefore, for timescales shorter than the radiation-reaction time $1/\eta$, the linear approximation in the spin is a valid approximation, which admits perturbative corrections in the spin.

Neglecting all $\mathcal{O}(\mathcal{S}^2)$ terms, Eq. (18) simply becomes $\mathfrak{m} = \mu$, which leads to the usual relation between the impulsion and the 4-velocity,

$$p^{\mu} = \mu v^{\mu}. \tag{54}$$

Once linearized in spin, the MPT equations (4)-(5) reduce to

$$\frac{Dp^{\mu}}{d\tau} = f_S^{(1)\mu} \triangleq -\frac{1}{2\mu}R^{\mu}{}_{\nu\alpha\beta}p^{\nu}S^{\alpha\beta}, \tag{55}$$

$$\frac{DS^{\mu}}{d\tau} = 0, \tag{56}$$

which are, respectively, the forced geodesic equation with force $f_S^{(1)\mu} = \mathcal{O}(\mathcal{S}^1)$ and the parallel transport equation of the spin vector studied e.g. in [27, 37].

## III. BUILDING CONSERVED QUANTITIES

The MPT equations take the form of a set of first order partial differential equations for the impulsions $p^{\mu}$ and the spin $S^{\mu}$. However, the very goal of anyone wanting to solve the MPT equations is to obtain the position of the spinning test-particle as a function of the proper time $X^{\mu}(\tau)$. With respect to the positions $X^{\mu}$, the MPT equations are a set of *second order* PDEs. As it is always the case when studying such a dynamical system, much information can be obtained if one is able to build *first integrals* of the motion (or invariants or conserved quantities), *i.e.* functions of the dynamical variables $\mathcal{Q}(p^{\alpha}, S^{\alpha})$ that are constant along the motion:

$$\dot{\mathcal{Q}}(p^{\alpha}, S^{\alpha}) = 0, \qquad \cdot \triangleq \frac{d}{d\tau}. \tag{57}$$

As always in General Relativity, the existence of first integrals of the motion will be strongly related with the presence of symmetries of the background spacetime (i.e. the existence of Killing vectors or Killing(-Yano) tensors).

In this Section, we will discuss the construction of invariants for the MPT equations that are at most linear in the spin vector. After a review of the conserved quantities already discussed in the literature, we will formulate the problem of finding an invariant (of the nonlinear MPT equations) at most linear in the spin as a set

of general constraints, following the method introduced by Rüdiger [22, 23]. The ideas behind this procedure are conceptually simple, but the computations for the MPT equations turn out to be involved. This is the reason why we will open this section by explaining Rüdiger's method for the case of the geodesic equations.

### A. The geodesic case

As a warm-up, let us recall how first integrals can be constructed for geodesic motion, namely when the spin-dipole is vanishing. In this case, the linear momentum is tangent to the worldline, $p^\mu = \mu v^\mu$ and the MPT equations reduce to the geodesic ones:

$$\frac{Dp^\mu}{d\tau} = 0, \qquad \frac{D}{d\tau} \triangleq v^\alpha \nabla_\alpha. \qquad (58)$$

In most GR textbooks and lectures, the problem is tackled from the perspective "*symmetry implies conservation*": one first introduces the notion of Killing vector fields ($\nabla_{(\alpha}\xi_{\beta)} = 0$) and *then* prove the well-known property stating that, given any geodesic of 4-impulsion $p^\mu$ and a Killing vector field $\xi^\mu$ of the background space-time, the quantity $C_\xi \triangleq \xi_\alpha p^\alpha$ is constant along the geodesic. One also shows that this property generalizes in the presence of a Killing tensor, and that the invariant mass $\mu^2 = -p_\alpha p^\alpha$ is also constant along the geodesic trajectory. Of course, when we assert that a scalar function $C(p^\alpha)$ is constant along the geodesic motion (or is conserved, or is invariant, or...), we have in mind the statement that it remains constant along the proper time evolution, which is equivalent to the statement that its covariant derivative along the geodesic path vanishes, namely

$$\dot{C}(p^\alpha) = 0 \quad \Leftrightarrow \quad p^\mu \nabla_\mu C(p^\alpha) = 0. \qquad (59)$$

In what follows, and as a prelude for the MPT case, we will tackle the problem in the opposite way ("*conservation requires symmetry*"): given an arbitrary geodesic, is it possible to construct invariants of the motion that are polynomial quantities of the impulsion, *i.e.* that are composed of monomials of the form

$$C_{\mathbf{K}}^{(n)} \triangleq K_{\alpha_1...\alpha_n} p^{\alpha_1} \ldots p^{\alpha_n} \qquad (60)$$

where, at this point, $\mathbf{K}$ is an arbitrary, by definition totally symmetric tensor of rank $n$? We will show that requiring the conservation of $C_{\mathbf{K}}^{(n)}$ will require either $\mathbf{K}$ to be a Killing vector/tensor or either that $C_{\mathbf{K}}^{(2)}$ is the invariant mass.

In order to work out the most general constraint on $\mathbf{K}$, we plug the definition of $C_{\mathbf{K}}$ (60) into the conservation equation (59). Using the geodesic equation (58) and relabelling the indices, one gets

$$p^\mu \nabla_\mu K_{\alpha_1...\alpha_n} p^{\alpha_1} \ldots p^{\alpha_n} = 0. \qquad (61)$$

The crucial point is that the dynamical variables $p^\alpha$ are *independent* among themselves. The above relation must hold for any values of the independent $p^\alpha$, yielding the general constraint

$$\nabla_{(\mu} K_{\alpha_1...\alpha_n)} = 0. \qquad (62)$$

The only possible cases for solving this constraint are the following:

- for $n = 1$, $K_\mu$ must be a Killing vector, $\nabla_{(\mu} K_{\nu)} = 0$;

- for $n = 2$, either $K_{\mu\nu}$ must be a rank-2 Killing tensor ($\nabla_{(\mu} K_{\nu\rho)} = 0$), either one takes $K_{\mu\nu} = g_{\mu\nu}$ which leads to the conservation of the invariant mass, $C_{\mathbf{g}}^{(2)} = -\mu^2$;

- for any $n \geq 3$, $\mathbf{K}$ must be a rank-n Killing tensor.

Before turning to the spinning particle case, let us make a couple of remarks:

1. As stated above, the viewpoint adopted here is reversed with respect to the 'traditional' one: we have proven that the existence of conserverd quantities along geodesic trajectories that are *polynomial* in the impulsions require the existence of symmetries of the background spacetime (except for the invariant mass $\mu$ which is always conserved).

2. Any linear combination of the invariants defined above remains of course invariant. Nevertheless, the conservation can be checked separately at each order in $\mathbf{p}$ because the application of the conservation condition (59) doesn't change the order in $\mathbf{p}$ of the terms contained in the resulting expression.

3. The invariant related to a Killing tensor is relevant only if the latter is irreducible, *i.e.* if it cannot be written as the product of Killing vectors. Otherwise, the invariant at order $n$ in $\mathbf{p}$ is just a product of invariants of lower order.

### B. The spinning case

We will now turn on the spin, and discuss the invariants that can be build for MPT equations.

#### 1. State-of-the-art

Several conserved quantities for the MPT equations have been discussed in the literature, most of the time in the context of the background spacetime being given by the Kerr metric [38]:

- When the Tulczyjew SSCs are enforced, the invariant mass

$$\mu^2 = -p_\alpha p^\alpha \tag{63}$$

and the spin parameter

$$S = \sqrt{S_\alpha S^\alpha} \tag{64}$$

are constant of the motion, as asserted by Eqs. (19) and (20).

- If there exists a Killing vector field $\xi_\alpha$ of the background spacetime, one can upgrade the geodesic case construction and show that

$$\mathcal{C}_\xi \triangleq \xi_\mu p^\mu + \frac{1}{2}\nabla_\mu \xi_\nu S^{\mu\nu} = C_\xi + \frac{1}{2}\nabla_\mu \xi_\nu S^{\mu\nu} \tag{65}$$

is an invariant of the motion [39]. It naturally arises from generalized Killing equations [40].

These are the only "obvious" invariants. To go further, one should apply the general procedure described above: write down the most general expression for the invariant we are looking for and then work out the constraints implied by its conservation along the motion.

In the presence of both $p^\mu$ and $S^\mu$, one can introduce a grading such that $[p^\mu] = [S^\mu] = [\mu] = [S^{\mu\nu}] = 1$ and $[g_{\mu\nu}] = 0$ and consider invariants of order $n$ along such grading ($n = 1$ linear, $n = 2$ quadratic, etc). Such a general scheme has been undertaken by R. Rüdiger in the early 80's, both for linear invariants [22] and for quadratic ones [23]:

- Restricting to invariants linear in $p^\mu$, $S^\mu$, Rüdiger found that, if there exists a Killing-Yano (KY) tensor $Y_{\alpha\beta}$ of the background spacetime, then the quantity

$$\mathcal{Q}_\mathbf{Y} \triangleq S_{\alpha\beta} {}^*Y^{\alpha\beta} \tag{66}$$

is a *quasi-invariant* of motion (i.e. conserved at linear order in the spin),

$$\dot{\mathcal{Q}}_\mathbf{Y} = \mathcal{O}(\mathcal{S}^2). \tag{67}$$

It would be an invariant of the motion at all orders in the spin provided that $\mathbf{Y}$ satisfy a set of additional conditions [22] which have been shown by

Santos and Batista [24] to be equivalent to requiring that $\mathbf{Y}$ is a covariantly constant Killing-Yano tensor on the background spacetime, which rules out the Kerr background. However, no systematic analysis has been performed yet on deforming the invariant $\mathcal{Q}_\mathbf{Y}$ with quadratic corrections in order to attempt to construct quasi-invariants at quadratic or higher order while not further restricting the background.

In this work, we will simply consider the quasi-invariant $\mathcal{Q}_\mathbf{Y}$ as one non-trivial quasi-conserved quantity relevant for the description of EMRIs.

- Similarly, restricting to quadratic order, Rüdiger showed that the quantity

$$\mathcal{Q}_R = -L_\alpha L^\alpha - 2\mu S^\alpha \partial_\alpha \mathcal{Z} - 2\mu^{-1} L_\alpha S^\alpha \xi_\beta p^\beta \tag{68}$$

where $L_\alpha \triangleq Y_{\alpha\lambda} p^\lambda$, $\mathcal{Z} \triangleq \frac{1}{4}Y^*_{\alpha\beta}Y^{\alpha\beta}$ and $\xi^\alpha \triangleq -\frac{1}{3}\nabla_\lambda Y^{*\lambda\alpha}$ is also a quasi-invariant, *i.e.* $\dot{\mathcal{Q}}_R = \mathcal{O}(\mathcal{S}^2)$, if $\mathbf{Y}$ is a Killing-Yano tensor.

Both non-trivial quasi-invariants appeared in the independent Hamilton-Jacobi method treatment of the equations of motion by Witzany [41], which illustrates the relevance of these quasi-invariants for the description of motion.

There is however not yet a proof that Rüdiger's quadratic invariant $\mathcal{Q}_R$ is the unique solution to the corresponding set of constraints in the presence of a Killing-Yano tensor. In what follows, we aim to (i) work out and simplify as much as possible Rüdiger's constraint equations; (ii) characterize the algebraic properties to be solved for obtaining the general solution and (iii) discuss the existence of additional quasi-invariants for the linearized MPT equations in a Kerr background.

### 2. General set of constraints for a quadratic invariant

Our principal motivation being the study of the *linearized* MPT equations, let us consider a quadratic invariant that is at most linear in $\mathbf{S}$:

$$\mathcal{Q} \triangleq K_{\mu\nu} p^\mu p^\nu + L_{\mu\nu\rho} S^{\mu\nu} p^\rho. \tag{69}$$

The tensors $\mathbf{K}$ and $\mathbf{L}$, which are by definition independent of $\mathbf{p}$ and $\mathbf{S}$, satisfy the algebraic symmetries $K_{\mu\nu} = K_{(\mu\nu)}$ and $L_{\mu\nu\rho} = L_{[\mu\nu]\rho}$.

Using the MPT equations (4)-(5) and the expression (15) for the 4-velocity, the conservation equation can be written

$$\dot{\mathcal{Q}} = \Xi\left(p^\lambda + \frac{1}{\mu}{}^*R^{*\lambda}{}_{\kappa\theta\sigma}S^\kappa S^\theta \hat{p}^\sigma\right)$$
$$\times \left[\left(\nabla_\lambda K_{\mu\nu} - 2L_{\lambda\mu\nu}\right)p^\mu p^\nu + \left(\nabla_\lambda L_{\alpha\beta\mu} - K_{\mu\rho}R^\rho{}_{\lambda\alpha\beta}\right)S^{\alpha\beta}p^\mu - \frac{1}{2}L_{\alpha\beta\rho}R^\rho{}_{\lambda\gamma\delta}S^{\alpha\beta}S^{\gamma\delta}\right] \overset{!}{=} 0, \tag{70}$$

with $\dot{Q} \triangleq \frac{d}{d\tau}Q$ and where the coefficient $\Xi \triangleq \frac{m^2}{\mu^2(1-d/2)}$ is non-vanishing. Let us introduce the tensors

$$U_{\alpha\beta\gamma} \triangleq \nabla_\gamma K_{\alpha\beta} - 2L_{\gamma(\alpha\beta)}, \tag{71}$$

$$V_{\alpha\beta\gamma\delta} \triangleq \nabla_\delta L_{\alpha\beta\gamma} - K_{\lambda\gamma}R^\lambda{}_{\delta\alpha\beta} + \frac{2}{3}K_{\lambda\rho}R^\lambda{}_{\delta[\alpha}{}^\rho g_{\beta]\gamma}, \tag{72}$$

$$W_{\alpha\beta\gamma\delta\varepsilon} \triangleq -\frac{1}{2}L_{\alpha\beta\lambda}R^\lambda{}_{\gamma\delta\varepsilon}. \tag{73}$$

The tensors $\mathbf{U}$, $\mathbf{V}$ and $\mathbf{W}$ obey the algebraic symmetries $U_{\alpha\beta\gamma} = U_{(\alpha\beta)\gamma}$, $V_{\alpha\beta\gamma\delta} = V_{[\alpha\beta]\gamma\delta}$, $W_{\alpha\beta\gamma\delta\varepsilon} = W_{[\alpha\beta]\gamma\delta\varepsilon}$. Notice that the orthogonality conditions $S^{\alpha\beta}p_\beta = 0$ imply

$$\frac{2}{3}K_{\lambda\rho}R^\lambda{}_{\delta[\alpha}{}^\rho g_{\beta]\gamma}S^{\alpha\beta}p^\gamma = 0. \tag{74}$$

The conservation equation reads as

$$\dot{Q} = \Xi\left(p^\lambda + \frac{1}{\mu}{}^*R^{*\lambda}{}_{\kappa\theta\sigma}S^\kappa S^\theta \hat{p}^\sigma\right)\left[U_{\mu\nu\lambda}p^\mu p^\nu + V_{\alpha\beta\mu\lambda}S^{\alpha\beta}p^\mu + W_{\alpha\beta\lambda\gamma\delta}S^{\alpha\beta}S^{\gamma\delta}\right] \overset{!}{=} 0. \tag{75}$$

We will go through a number of steps in order to express this condition in terms of the independent variables $s_\alpha$ and $\hat{p}^\mu$. First, let us expand all terms and express the spin-related quantities in terms of the independent variables $s^\alpha$. For this purpose, we will make use of the identities

$$p^{[\alpha}S^{\beta]} = p^{[\alpha}s^{\beta]}, \qquad S^{\alpha\beta} = 2S^{[\alpha}\hat{p}^{\beta]*} = 2s^{[\alpha}\hat{p}^{\beta]*}, \qquad S^\alpha = \Pi^\alpha_\beta s^\beta. \tag{76}$$

The conservation equation becomes

$$\dot{Q} = \frac{\Xi}{\mu}\left[\mu^4 U_{\mu\nu\rho}\hat{p}^\mu\hat{p}^\nu\hat{p}^\rho + 2\mu^3{}^*V_{\alpha\mu\nu\rho}s^\alpha\hat{p}^\mu\hat{p}^\nu\hat{p}^\rho + \mu^2\left(4{}^*W^*{}_{\alpha\mu\nu\beta\rho} + {}^*R^{*\lambda}{}_{\kappa\alpha\rho}U_{\mu\nu\lambda}\Pi^\kappa_\beta\right)s^\alpha s^\beta\hat{p}^\mu\hat{p}^\nu\hat{p}^\rho\right.$$

$$\left. + 2\mu{}^*R^{*\lambda}{}_{\kappa\alpha\rho}{}^*V_{\gamma\mu\nu\lambda}\Pi^\kappa_\beta s^\alpha s^\beta s^\gamma\hat{p}^\mu\hat{p}^\nu\hat{p}^\rho + 4{}^*R^{*\lambda}{}_{\kappa\alpha\rho}{}^*W^*{}_{\delta\mu\lambda\gamma\nu}\Pi^\kappa_\beta s^\alpha s^\beta s^\gamma s^\delta\hat{p}^\mu\hat{p}^\nu\hat{p}^\rho\right] \overset{!}{=} 0. \tag{77}$$

Second, we will remove the projectors. One has the identity

$${}^*R^{*\lambda}{}_{\kappa\alpha\rho}\Pi^\kappa_\beta s^\alpha s^\beta = -I^{\lambda\alpha\beta}{}_{\rho\sigma\kappa}s_\alpha s_\beta\hat{p}^\sigma\hat{p}^\kappa \tag{78}$$

where we have defined

$$I^{\lambda\alpha\beta}{}_{\rho\sigma\kappa} \triangleq {}^*R^{*\lambda}{}_{\kappa\rho}{}^\alpha\delta^\beta_\sigma + {}^*R^{*\lambda\alpha\beta}{}_\rho g_{\sigma\kappa}. \tag{79}$$

The proof is easily carried out, using the fact that $\hat{p}_\mu\hat{p}^\mu = -1$:

$${}^*R^{*\lambda}{}_{\kappa\alpha\rho}\Pi^\kappa_\beta s^\alpha s^\beta = {}^*R^{*\lambda}{}_{\kappa\alpha\rho}\left(\delta^\kappa_\beta + \hat{p}^\kappa\hat{p}_\beta\right)s^\alpha s^\beta \tag{80}$$

$$= \left[(-\hat{p}^\sigma\hat{p}^\kappa g_{\sigma\kappa}){}^*R^{*\lambda}{}_{\beta\alpha\rho} + {}^*R^{*\lambda}{}_{\kappa\alpha\rho}\hat{p}^\kappa\delta^\sigma_\beta\hat{p}_\sigma\right]s^\alpha s^\beta \tag{81}$$

$$= -\left({}^*R^{*\lambda}{}_{\kappa\rho}{}^\alpha\delta^\beta_\sigma + {}^*R^{*\lambda\alpha\beta}{}_\rho g_{\sigma\kappa}\right)s_\alpha s_\beta\hat{p}^\sigma\hat{p}^\kappa. \tag{82}$$

Using this identity, the conservation equation finally reads as

$$\dot{Q} = \frac{\Xi}{\mu}\left[\mu^4 U_{\mu\nu\rho}\hat{p}^\mu\hat{p}^\nu\hat{p}^\rho + 2\mu^3{}^*V^\alpha{}_{\mu\nu\rho}s_\alpha\hat{p}^\mu\hat{p}^\nu\hat{p}^\rho - \mu^2\left(I^{\lambda\alpha\beta}{}_{\rho\sigma\kappa}U_{\mu\nu\lambda} + 4{}^*W^{*\alpha}{}_{\mu\nu}{}^\beta{}_\rho g_{\sigma\kappa}\right)s_\alpha s_\beta\hat{p}^\mu\hat{p}^\nu\hat{p}^\rho\hat{p}^\sigma\hat{p}^\kappa\right.$$

$$\left. - 2\mu I^{\lambda\alpha\beta}{}_{\rho\sigma\kappa}{}^*V^\gamma{}_{\mu\nu\lambda}s_\alpha s_\beta s_\gamma\hat{p}^\mu\hat{p}^\nu\hat{p}^\rho\hat{p}^\sigma\hat{p}^\kappa - 4I^{\lambda\alpha\beta}{}_{\rho\sigma\kappa}{}^*W^{*\gamma}{}_{\mu\lambda}{}^\delta{}_\nu s_\alpha s_\beta s_\gamma s_\delta\hat{p}^\mu\hat{p}^\nu\hat{p}^\rho\hat{p}^\sigma\hat{p}^\kappa\right] \overset{!}{=} 0. \tag{83}$$

In principle, finding a quantity which is *exactly* con-    served along the motion generated by the MPT equa-

tions requires condition (83) to be satisfied *exactly*. Because the variables $\hat{p}^\mu$ and $s_\alpha$ are independent, this requirement is equivalent to the following set of five constraints, each of them arising at a different order in the spin parameter:

$$\mathcal{O}(\mathcal{S}^0) : U_{(\mu\nu\rho)} = 0, \tag{84}$$

$$\mathcal{O}(\mathcal{S}^1) : {}^*V^\alpha{}_{(\mu\nu\rho)} = 0, \tag{85}$$

$$\mathcal{O}(\mathcal{S}^2) : I^{\lambda(\alpha\beta}{}_{(\mu\nu\rho}U_{\sigma\kappa)\lambda} + 4{}^*W^{*(\alpha}{}_{(\mu\nu}{}^{\beta)}{}_\rho g_{\sigma\kappa)} = 0, \tag{86}$$

$$\mathcal{O}(\mathcal{S}^3) : I^{\lambda(\alpha\beta}{}_{(\mu\nu\rho}{}^*V^{\gamma)}{}_{\sigma\kappa)\lambda} = 0, \tag{87}$$

$$\mathcal{O}(\mathcal{S}^4) : I^{\lambda(\alpha\beta}{}_{(\mu\nu\rho}{}^*W^{*\gamma}{}_{\sigma|\lambda|}{}^{\delta)}{}_{\kappa)} = 0. \tag{88}$$

However, for physically relevant situations (*e.g.* when the background spacetime is Kerr or Schwarschild), it will not be possible to fulfill all the constraints (84) to (88). Nevertheless, we will be able to work out an explicit solution to the two first constraints (84) and (85) for any Ricci-flat spacetime admitting a hidden symmetry encoded under the form of a rank-two Killing-Yano tensor. In this case, the conservation equation (83) will

*not* be satisfied exactly, but takes the form

$$\dot{\mathcal{Q}} = \mathcal{O}(\mathcal{S}^2), \tag{89}$$

*i.e.* $\mathcal{Q}$ is a quasi-invariant in the sense defined above.

Notice that the $\mathcal{O}(\mathcal{S}^0)$ constraint (84) simply reduces to

$$\nabla_{(\alpha}K_{\beta\gamma)} = 0, \tag{90}$$

*i.e.* **K** must be a Killing tensor of the background spacetime.

The $\mathcal{O}(\mathcal{S}^1)$ constraint (85) is more difficult to work out. In Section IV, we will proceed to a clever rewriting of this constraint, which will then be particularized to spacetimes admitting a Killing-Yano tensor in Section V. Section VI will aim to solve it generally. Finally, all these results will be particularized to a Kerr background in Section VII.

## IV. CONSERVATION EQUATION AT LINEAR ORDER IN THE SPIN

We will now proceed to the aforementioned rewriting of the constraint (85) by introducing a new set of variables. The three first parts of this section are devoted to the derivation of preliminary results, that will be crucial for working out the main result.

### A. Dual form of V

We want to compute the dual form of the tensor

$$V_{\alpha\beta\gamma\delta} \triangleq \nabla_\delta L_{\alpha\beta\gamma} - K_{\lambda\gamma}R^\lambda{}_{\delta\alpha\beta} + \frac{2}{3}K_{\lambda\rho}R^\lambda{}_{\delta[\alpha}{}^\rho g_{\beta]\gamma} \tag{91}$$

with respect to its two first indices. One has

$${}^*V_{\alpha\beta\gamma\delta} = \nabla_\delta {}^*L_{\alpha\beta\gamma} - K_{\lambda\gamma}R^{*\lambda}{}_{\delta\alpha\beta} + \frac{2}{3}K_{\lambda\rho}R^\lambda{}_{\delta[\alpha}{}^\rho g_{\beta]*\gamma}. \tag{92}$$

The last term of this equality can be written as

$$\frac{2}{3}K_{\lambda\rho}R^\lambda{}_{\delta[\alpha}{}^\rho g_{\beta]*\gamma} = \frac{1}{3}K^{\lambda\rho}\varepsilon_{\alpha\beta}{}^{\mu\nu}R_{\lambda\delta\mu\rho}g_{\nu\gamma} = \frac{1}{3}K^{\lambda\rho}\varepsilon_{\alpha\beta}{}^\mu{}_\gamma R_{\lambda\delta\mu\rho} \tag{93}$$

$$= \frac{1}{3}K^{\lambda[\mu}\varepsilon_{\alpha\beta}{}^{\nu]}{}_\gamma R^{**}_{\lambda\delta\mu\nu} = \frac{1}{3}K^{\lambda[\mu}\varepsilon_{\alpha\beta}{}^{\nu]*}{}_\gamma R^*_{\lambda\delta\mu\nu} \tag{94}$$

$$= -\frac{1}{6}\varepsilon^{\sigma\mu\nu\rho}\varepsilon_{\sigma\alpha\beta\gamma}K_{\lambda\rho}R^{*\lambda}{}_{\delta\mu\nu} \tag{95}$$

$$= \frac{1}{3}\left(K_{\lambda\gamma}R^{*\lambda}{}_{\delta\alpha\beta} + K_{\lambda\alpha}R^{*\lambda}{}_{\delta\beta\gamma} + K_{\lambda\beta}R^{*\lambda}{}_{\delta\gamma\alpha}\right) \tag{96}$$

$$= \frac{1}{3}K_{\gamma\lambda}R^{*\lambda}{}_{\delta\alpha\beta} + \frac{2}{3}R^{*\lambda}{}_{\delta\gamma[\alpha}K_{\beta]\lambda}. \tag{97}$$

This finally yields

$$\boxed{{}^*V_{\alpha\beta\gamma\delta} = \nabla_\delta {}^*L_{\alpha\beta\gamma} - \frac{2}{3}K_{\lambda\gamma}R^{*\lambda}{}_{\delta\alpha\beta} + \frac{2}{3}R^{*\lambda}{}_{\delta\gamma[\alpha}K_{\beta]\lambda}.} \tag{98}$$

## B. Rüdiger variables

Following Rüdiger [23], let us introduce

$$\tilde{X}_{\alpha\beta\gamma} \triangleq L_{\alpha\beta\gamma} - \frac{1}{3}\left(\lambda_{\alpha\beta\gamma} + g_{\gamma[\alpha}\nabla_{\beta]}K\right), \tag{99}$$

where we have made use of the notations

$$\lambda_{\alpha\beta\gamma} \triangleq 2\nabla_{[\alpha}K_{\beta]\gamma}, \qquad K \triangleq K^{\alpha}{}_{\alpha}. \tag{100}$$

The irreducible parts $X_{\alpha}$ and $X_{\alpha\beta\gamma}$ of $\tilde{X}_{\alpha\beta\gamma}$ are defined through the relation

$$\tilde{X}_{\alpha\beta\gamma} \triangleq X_{\alpha\beta\gamma} + \varepsilon_{\alpha\beta\gamma\delta}X^{\delta}, \qquad \text{with } X_{[\alpha\beta\gamma]} \stackrel{!}{=} 0. \tag{101}$$

They provide an equivalent description, since Eq. (101) can be inverted as

$$X_{\alpha\beta\gamma} = \tilde{X}_{\alpha\beta\gamma} - \tilde{X}_{[\alpha\beta\gamma]}, \tag{102}$$

$$X^{\alpha} = \frac{1}{6}\varepsilon^{\alpha\beta\gamma\delta}\tilde{X}_{\beta\gamma\delta}. \tag{103}$$

Finally, a simple computation shows that the dual of $\tilde{\mathbf{X}}$ is given by

$${}^{*}\tilde{X}_{\alpha\beta\gamma} = {}^{*}X_{\alpha\beta\gamma} - 2g_{\gamma[\alpha}X_{\beta]}. \tag{104}$$

## C. The structural equation

This third preliminary part will be devoted to the proof of the *structural equation* [23]

$$\boxed{\nabla_{\delta}\lambda_{\alpha\beta\gamma} = 2\left(R^{\lambda}{}_{\delta\alpha\beta}K_{\gamma\lambda} - R^{\lambda}{}_{\delta\gamma[\alpha}K_{\beta]\lambda}\right) + \mu_{\alpha\beta\gamma\delta}} \tag{105}$$

with

$$\boxed{\mu_{\alpha\beta\gamma\delta} \triangleq \frac{1}{2}\left[K_{\beta\gamma;(\alpha\delta)} + K_{\alpha\delta;(\beta\gamma)} - K_{\alpha\gamma;(\beta\delta)} - K_{\beta\delta;(\alpha\gamma)} - 3\left(K_{\lambda[\alpha}R^{\lambda}{}_{\beta]\gamma\delta} + K_{\lambda[\gamma}R^{\lambda}{}_{\delta]\alpha\beta}\right)\right].} \tag{106}$$

$\mu$ possesses the same algebraic symmetries than the Riemann tensor. We remind the reader that $\lambda_{\alpha\beta\gamma} \triangleq 2\nabla_{[\alpha}K_{\beta]\gamma}$. We will use indifferently the notations $\nabla_{\alpha}\mathbf{T}$ or $\mathbf{T}_{;\alpha}$ for the covariant derivative of the tensor $\mathbf{T}$. The proof goes as a lengthy rewriting of the original expression:

$$\nabla_{\delta}\lambda_{\alpha\beta\gamma} = \nabla_{\delta}\nabla_{\alpha}K_{\beta\gamma} - \nabla_{\delta}\nabla_{\beta}K_{\alpha\gamma} \tag{107}$$

$$= \nabla_{(\delta}\nabla_{\alpha)}K_{\beta\gamma} + \nabla_{[\delta}\nabla_{\alpha]}K_{\beta\gamma} - \nabla_{(\delta}\nabla_{\beta)}K_{\alpha\gamma} - \nabla_{[\delta}\nabla_{\beta]}K_{\alpha\gamma} \tag{108}$$

$$= \frac{1}{2}\nabla_{(\alpha}\nabla_{\delta)}K_{\beta\gamma} - \frac{1}{2}\nabla_{(\beta}\nabla_{\delta)}K_{\alpha\gamma} + \frac{1}{2}\left(\nabla_{(\alpha}\nabla_{\delta)}K_{\beta\gamma} - \nabla_{(\beta}\nabla_{\delta)}K_{\alpha\gamma}\right) + \frac{1}{2}[\nabla_{\delta},\nabla_{\alpha}]K_{\beta\gamma} - \frac{1}{2}[\nabla_{\delta},\nabla_{\beta}]K_{\alpha\gamma}. \tag{109}$$

We proceed to the following rewriting of twice the quantity in brackets contained in the above expression:

$$\nabla_{\alpha}\nabla_{\delta}K_{\beta\gamma} + \nabla_{\delta}\nabla_{\alpha}K_{\beta\gamma} - \nabla_{\beta}\nabla_{\delta}K_{\alpha\gamma} - \nabla_{\delta}\nabla_{\beta}K_{\alpha\gamma} \tag{110}$$

$$= 2\nabla_{\alpha}\nabla_{\delta}K_{\beta\gamma} - 2\nabla_{\beta}\nabla_{\delta}K_{\alpha\gamma} + [\nabla_{\delta},\nabla_{\alpha}]K_{\beta\gamma} - [\nabla_{\delta},\nabla_{\beta}]K_{\alpha\gamma} \tag{111}$$

$$= 2\left(\nabla_{\beta}\nabla_{\alpha}K_{\gamma\delta} + \nabla_{\beta}\nabla_{\gamma}K_{\alpha\delta} - \nabla_{\alpha}\nabla_{\beta}K_{\gamma\delta} - \nabla_{\alpha}\nabla_{\gamma}K_{\beta\delta}\right) + [\nabla_{\delta},\nabla_{\alpha}]K_{\beta\gamma} - [\nabla_{\delta},\nabla_{\beta}]K_{\alpha\gamma} \tag{112}$$

$$= 2\left(\nabla_{(\beta}\nabla_{\gamma)}K_{\alpha\delta} - \nabla_{(\alpha}\nabla_{\gamma)}K_{\beta\delta}\right) + [\nabla_{\delta},\nabla_{\alpha}]K_{\beta\gamma} - [\nabla_{\delta},\nabla_{\beta}]K_{\alpha\gamma}$$
$$\quad - 2[\nabla_{\alpha},\nabla_{\beta}]K_{\gamma\delta} - [\nabla_{\alpha},\nabla_{\gamma}]K_{\beta\delta} + [\nabla_{\beta},\nabla_{\gamma}]K_{\alpha\delta}. \tag{113}$$

This yields

$$\nabla_{\delta}\lambda_{\alpha\beta\gamma} = \frac{1}{2}\left(K_{\beta\gamma;(\alpha\delta)} + K_{\alpha\delta;(\beta\gamma)} - K_{\alpha\gamma;(\beta\delta)} - K_{\beta\delta;(\alpha\gamma)}\right)$$
$$\quad + \underbrace{\frac{3}{4}[\nabla_{\delta},\nabla_{\alpha}]K_{\beta\gamma} - \frac{3}{4}[\nabla_{\delta},\nabla_{\beta}]K_{\alpha\gamma} - \frac{1}{2}[\nabla_{\alpha},\nabla_{\beta}]K_{\gamma\delta} - \frac{1}{4}[\nabla_{\alpha},\nabla_{\gamma}]K_{\beta\delta} + \frac{1}{4}[\nabla_{\beta},\nabla_{\gamma}]K_{\alpha\delta}}_{\triangleq\heartsuit}. \tag{114}$$

The quantity $\heartsuit$ can be rearranged in the following way:

$$4\heartsuit = 3[\nabla_\delta, \nabla_\alpha]K_{\beta\gamma} - 3[\nabla_\delta, \nabla_\beta]K_{\alpha\gamma} - 2[\nabla_\alpha, \nabla_\beta]K_{\gamma\delta} - [\nabla_\alpha, \nabla_\gamma]K_{\beta\delta} + [\nabla_\beta, \nabla_\gamma]K_{\alpha\delta} \tag{115}$$

$$= \left(3R^\lambda{}_{\gamma\delta\beta} - R^\lambda{}_{\delta\beta\gamma}\right)K_{\alpha\lambda} + \left(R^\lambda{}_{\delta\alpha\gamma} - 3R^\lambda{}_{\gamma\delta\alpha}\right)K_{\beta\lambda}$$

$$+ \left(3R^\lambda{}_{\alpha\delta\beta} - 3R^\lambda{}_{\beta\delta\alpha} + 2R^\lambda{}_{\delta\alpha\beta}\right)K_{\gamma\lambda} + \left(2R^\lambda{}_{\gamma\alpha\beta} + R^\lambda{}_{\beta\alpha\gamma} - R^\lambda{}_{\alpha\beta\gamma}\right)K_{\delta\lambda} \tag{116}$$

$$= 5R^\lambda{}_{\delta\alpha\beta}K_{\gamma\lambda} + 4\left(R^\lambda{}_{\delta\alpha\gamma}K_{\beta\lambda} - R^\lambda{}_{\delta\beta\gamma}K_{\alpha\lambda}\right) + 3\left(R^\lambda{}_{\alpha\gamma\delta}K_{\beta\lambda} - R^\lambda{}_{\beta\gamma\delta}K_{\alpha\lambda} + R^\lambda{}_{\gamma\alpha\beta}K_{\delta\lambda}\right) \tag{117}$$

$$= 5R^\lambda{}_{\delta\alpha\beta}K_{\gamma\lambda} + 8R^\lambda{}_{\delta[\alpha|\gamma}K_{|\beta]\lambda} - 6K_{\lambda[\alpha}R^\lambda{}_{\beta]\gamma\delta} + 3R^\lambda{}_{\gamma\alpha\beta}K_{\delta\lambda} \underbrace{-3R^\lambda{}_{\delta\alpha\beta}K_{\gamma\lambda} + 3R^\lambda{}_{\delta\alpha\beta}K_{\gamma\lambda}}_{=0} \tag{118}$$

$$= 8R^\lambda{}_{\delta\alpha\beta}K_{\gamma\lambda} - 8R^\lambda{}_{\delta\gamma[\alpha}K_{\beta]\lambda} - 6K_{\lambda[\alpha}R^\lambda{}_{\beta]\gamma\delta} - 6K_{\lambda[\gamma}R^\lambda{}_{\delta]\alpha\beta}. \tag{119}$$

Consequently,

$$\heartsuit = 2\left(R^\lambda{}_{\delta\alpha\beta}K_{\gamma\lambda} - R^\lambda{}_{\delta\gamma[\alpha}K_{\beta]\lambda}\right) - \frac{3}{2}\left(K_{\lambda[\alpha}R^\lambda{}_{\beta]\gamma\delta} + K_{\lambda[\gamma}R^\lambda{}_{\delta]\alpha\beta}\right). \tag{120}$$

Inserting this result into Eq. (114) gives the structural equation (105) and consequently concludes the proof.
We will conclude this section by working out the dual form of the structural equation (105). One has

$$\nabla_\delta {}^*\lambda_{\alpha\beta\gamma} = 2R^{*\lambda}{}_{\delta\alpha\beta}K_{\gamma\lambda} - \varepsilon_{\alpha\beta}{}^{\mu\nu}R^\lambda{}_{\delta\gamma\mu}K_{\nu\lambda} + {}^*\mu_{\alpha\beta\gamma\delta} \tag{121}$$

$$= 2R^{*\lambda}{}_{\delta\alpha\beta}K_{\gamma\lambda} + \varepsilon_{\alpha\beta}{}^{\mu\nu}R^{**\lambda}{}_{\delta\gamma\mu}K_{\nu\lambda} + {}^*\mu_{\alpha\beta\gamma\delta}. \tag{122}$$

It is now easier to compute

$$\nabla_\delta {}^*\lambda_{\alpha\beta}{}^\gamma = 2R^{*\lambda}{}_{\delta\alpha\beta}K^\gamma{}_\lambda - \frac{1}{2}\varepsilon_{\mu\alpha\beta\nu}\varepsilon^{\mu\gamma\rho\sigma}R^{*\lambda}{}_{\delta\rho\sigma}K^\nu{}_\lambda + {}^*\mu_{\alpha\beta}{}^\gamma{}_\delta \tag{123}$$

$$= 2R^{*\lambda}{}_{\delta\alpha\beta}K^\gamma{}_\lambda + 3\delta_\alpha^{[\gamma}\delta_\beta^\rho\delta_\nu^{\sigma]}R^{*\lambda}{}_{\delta\rho\sigma}K^\nu{}_\lambda + {}^*\mu_{\alpha\beta}{}^\gamma{}_\delta, \tag{124}$$

which yields

$$\nabla_\delta {}^*\lambda_{\alpha\beta\gamma} = 2R^{*\lambda}{}_{\delta\alpha\beta}K_{\gamma\lambda} + 3g_{\gamma[\alpha|}R^{*\lambda}{}_{\delta|\beta\nu]}K^\nu{}_\lambda + {}^*\mu_{\alpha\beta\gamma\delta} \tag{125}$$

$$= 2R^{*\lambda}{}_{\delta\alpha\beta}K_{\gamma\lambda} + \left(R^{*\lambda}{}_{\delta\alpha\beta}K_{\gamma\lambda} + R^{*\lambda}{}_{\delta\beta}{}^\rho K_{\lambda\rho}g_{\alpha\gamma} - R^{*\lambda}{}_{\delta\alpha}{}^\rho K_{\lambda\rho}g_{\beta\gamma}\right) + {}^*\mu_{\alpha\beta\gamma\delta}. \tag{126}$$

Rearranging the different terms leads to the final expression

$$\boxed{\nabla_\delta {}^*\lambda_{\alpha\beta\gamma} = 3R^{*\lambda}{}_{\delta\alpha\beta}K_{\gamma\lambda} - 2R^{*\lambda}{}_{\delta[\alpha}{}^\rho g_{\beta]\gamma}K_{\lambda\rho} + {}^*\mu_{\alpha\beta\gamma\delta}.} \tag{127}$$

### D. Reduction of the second constraint

We will now gather the results obtained in the three previous subsections to express the constraint (85) in terms of the irreducible variables introduced above. Let us remind that Eq. (85) reads as

$$^*V^\alpha{}_{(\beta\gamma\delta)} = 0. \tag{128}$$

Using Eqs. (99), (104) and (127), we can rewrite

$$^*V_{\alpha\beta\gamma\delta} = \nabla_\delta {}^*\tilde{X}_{\alpha\beta\gamma} + \frac{1}{3}\nabla_\delta\left({}^*\lambda_{\alpha\beta\gamma} + g_{\gamma[\alpha}\nabla_{\beta]*}K\right) - \frac{2}{3}K_{\lambda\gamma}R^{*\lambda}{}_{\delta\alpha\beta} + \frac{2}{3}R^{*\lambda}{}_{\delta\gamma[\alpha}K_{\beta]\lambda} \tag{129}$$

$$= \nabla_\delta\left({}^*X_{\alpha\beta\gamma} - 2g_{\gamma[\alpha}X_{\beta]}\right) + \frac{1}{3}\underbrace{\left(R^{*\lambda}{}_{\delta\alpha\beta}K_{\gamma\lambda} + 2R^{*\lambda}{}_{\delta\gamma[\alpha}K_{\beta]\lambda}\right)}_{\triangleq\heartsuit}$$

$$+ \frac{1}{3}\nabla_\delta\left(g_{\gamma[\alpha}\nabla_{\beta]*}K\right) + \frac{1}{3}{}^*\mu_{\alpha\beta\gamma\delta} - \frac{2}{3}R^{*\lambda}{}_{\delta[\alpha}{}^\rho g_{\beta]\gamma}K_{\lambda\rho}. \tag{130}$$

On the one hand, we have

$$\heartsuit = R^{*\lambda}{}_{\delta\alpha\beta}K_{\gamma\lambda} + R^{*\lambda}{}_{\delta\gamma\alpha}K_{\beta\lambda} - R^{*\lambda}{}_{\delta\gamma\beta}K_{\alpha\lambda} = R^{*\lambda}{}_{\delta\beta\gamma}K_{\alpha\lambda} + 2R^{*\lambda}{}_{\delta\alpha[\beta}K_{\gamma]\lambda}. \tag{131}$$

And on the other hand, we can write

$$\nabla_\delta \left( g_{\gamma[\alpha}\nabla_{\beta]*}K \right) = \frac{1}{2}\varepsilon_{\alpha\beta}{}^{\mu\nu}g_{\gamma\mu}\nabla_\delta\nabla_\nu K = \frac{1}{2}\varepsilon_{\alpha\beta\gamma}{}^{\mu}\nabla_\delta\nabla_\mu K. \tag{132}$$

Gathering these expressions leads to

$$^*V_{\alpha\beta\gamma\delta} = \nabla_\delta{}^*X_{\alpha\beta\gamma} - 2g_{\gamma[\alpha|}\nabla_\delta X_{|\beta]} - \frac{2}{3}R^{*\lambda}{}_{\delta[\alpha}{}^{\rho}g_{\beta]\gamma}K_{\lambda\rho} + \frac{1}{3}R^{*\lambda}{}_{\delta\beta\gamma}K_{\alpha\lambda} + \frac{2}{3}R^{*\lambda}{}_{\delta\alpha[\beta}K_{\gamma]\lambda} + \frac{1}{6}\varepsilon_{\alpha\beta\gamma}{}^{\mu}\nabla_\delta\nabla_\mu K + \frac{1}{3}{}^*\mu_{\alpha\beta\gamma\delta}. \tag{133}$$

When symmetrizing the three last indices, the four last terms of this expression vanish. The constraint (85) takes the final form

$$\boxed{{}^*X_{\alpha(\beta\gamma;\delta)} + X_{\alpha;(\beta}g_{\gamma\delta)} - g_{\alpha(\beta}X_{\gamma;\delta)} + \frac{1}{3}\left( g_{\alpha(\beta}R^{*\lambda}{}_{\gamma\delta)}{}^{\rho} - R^{*\lambda}{}_{\alpha(\beta}{}^{\rho}g_{\gamma\delta)} \right)K_{\lambda\rho} = 0.} \tag{134}$$

Our principal motivation being the motion of spinning particles in Kerr spacetime, we will now focus on spacetimes possessing a Killing-Yano (KY) tensor. It will turn out that the constraint (134) can still be dramatically simplified in such a framework.

## V. SPACETIMES ADMITTING A KILLING-YANO TENSOR

We now particularize our analysis to spacetimes equipped with a Killing-Yano (KY) tensor, *i.e.* a rank-2, antisymmetric tensor $Y_{\mu\nu} = Y_{[\mu\nu]}$ obeying the Killing-Yano equation:

$$\boxed{\nabla_{(\alpha}Y_{\beta)\gamma} = 0.} \tag{135}$$

In this case, the constraint (84) is automatically fulfilled, because

$$K_{\alpha\beta} \triangleq Y_{\alpha\lambda}Y^{\lambda}{}_{\beta} \tag{136}$$

is a Killing tensor.

The aim of this section is twofold. First, we will review some general properties of KY tensors useful for the continuation of our analysis. Even if most of them were previously mentioned in the literature [22, 42], the goal of the present exposition is to provide a self-contained summary of these results and of their derivations. Second, we will work out an involved identity that will become a cornerstone for solving the constraint (134). This so-called *central identity* is the generalization of a result mentioned by Rüdiger in [23]. Rüdiger only quickly sketched the proof of his identity, while we aim here to provide a more pedagogical derivation of this central result.

### A. Some general properties of KY tensors

Let us review some basic properties of KY tensors, sticking to our conventions and simplifying the notations used in the literature [22, 42].

#### 1. An equivalent form of the KY equation

The symmetries $\nabla_{(\alpha}Y_{\beta)\gamma} = \nabla_\alpha Y_{(\beta\gamma)} = 0$ ensure the quantity $\nabla_\alpha Y_{\beta\gamma}$ to be totally antisymmetric in its three indices. Consequently, there exists a vector $\boldsymbol{\xi}$ such that

$$\boxed{\nabla_\alpha Y_{\beta\gamma} = \varepsilon_{\alpha\beta\gamma\lambda}\xi^{\lambda}.} \tag{137}$$

The value of $\boldsymbol{\xi}$ can be found by contracting Eq. (137) with $\varepsilon^{\mu\alpha\beta\gamma}$ and making use of the contraction formula for the Levi-Civita tensor (23). We obtain

$$\boxed{\xi^{\alpha} = -\frac{1}{3}\nabla_\lambda Y^{*\lambda\alpha}.} \tag{138}$$

Notice that Eq. (137) with $\boldsymbol{\xi}$ given by Eq. (138) is totally equivalent to the Killing-Yano equation.

#### 2. Dual KY equation

Let us derive the equivalent to the Killing-Yano equation for the dual of the KY tensor $\mathbf{Y}^*$. One has

$$\nabla_\alpha Y^{*\beta\gamma} = \frac{1}{2}\varepsilon^{\beta\gamma\mu\nu}\nabla_\alpha Y_{\mu\nu} = \frac{1}{2}\varepsilon^{\mu\nu\beta\gamma}\varepsilon_{\mu\nu\alpha\lambda}\xi^{\lambda} \tag{139}$$

$$= -2\delta_\alpha^{[\beta}\delta_\lambda^{\gamma]}\xi^{\lambda} = -2\delta_\alpha^{[\beta}\xi^{\gamma]}, \tag{140}$$

leading to

$$\boxed{\nabla_\alpha Y^*_{\beta\gamma} = -2g_{\alpha[\beta}\xi_{\gamma]}.} \tag{141}$$

In turn, this leads to the conformal Killing-Yano equation for the dual tensor $\mathbf{Y}^*$.

### 3. Integrability conditions for the KY equation

We will work out some necessary conditions for the tensor **Y** to be a Killing-Yano tensor, *i.e.* relations that must hold for **Y** to satisfy the KY equation. We will refer to them as *integrability conditions* for the Killing-Yano equation.

We begin by proving some preliminary results:

**Lemma 1.** *One has*

$$\boxed{\nabla^\alpha \xi_\alpha = 0.} \tag{142}$$

*Proof.* We proceed by applying the Ricci identity to the expression:

$$\nabla^\alpha \xi_\alpha = -\frac{1}{3}\nabla_\alpha \nabla_\lambda Y^{*\lambda\alpha} = -\frac{1}{6}[\nabla_\alpha,\nabla_\lambda]Y^{*\lambda\alpha} \tag{143}$$

$$= \frac{1}{6}g^{\lambda\mu}g^{\alpha\nu}\left(R^\rho{}_{\mu\alpha\lambda}Y^*_{\rho\nu} + R^\rho{}_{\nu\alpha\lambda}Y^*_{\mu\rho}\right) \tag{144}$$

$$= \frac{1}{3}R^{\alpha\beta}Y^*_{\alpha\beta} = 0. \tag{145}$$

$\square$

**Lemma 2.** *For any antisymmetric tensor $A_{\alpha\beta} = A_{[\alpha\beta]}$, one has*

$$\boxed{R^\alpha{}_{\mu\nu}{}^\beta A^{\mu\nu} = R^{[\alpha}{}_{\mu\nu}{}^{\beta]}A^{\mu\nu}.} \tag{146}$$

*Proof.* The proof is straightforward:

$$R^\beta{}_{\mu\nu}{}^\alpha A^{\mu\nu} = R_\nu{}^{\alpha\beta}{}_\mu A^{\mu\nu} = R^\alpha{}_{\mu\nu}{}^\beta A^{\nu\mu} \tag{147}$$

$$= -R^\alpha{}_{\mu\nu}{}^\beta A^{\mu\nu}. \tag{148}$$

$\square$

**Lemma 3.** *For any antisymmetric tensor $A_{\alpha\beta} = A_{[\alpha\beta]}$, we have*

$$\boxed{A_{\mu\nu}R^\mu{}_{\alpha\beta}{}^\nu = -\frac{1}{2}A^{\mu\nu}R_{\alpha\beta\mu\nu}.} \tag{149}$$

*Proof.* The proof is again pretty simple:

$$A_{\mu\nu}R^\mu{}_{\alpha\beta}{}^\nu = A^{\mu\nu}\left(R_{\alpha\beta\nu\mu} + R_{\alpha\nu\mu\beta}\right) \tag{150}$$

$$= -A^{\mu\nu}R_{\alpha\beta\mu\nu} + A^{\mu\nu}R_{\nu\alpha\beta\mu}. \tag{151}$$

The conclusion is reached using Lemma 2. $\square$

We can now work out the first integrability condition. One has

$$\nabla_\alpha \xi_\beta = -\frac{1}{3}\nabla_\alpha \nabla_\lambda Y^{*\lambda}{}_\beta \tag{152}$$

$$= -\frac{1}{3}\nabla^\lambda \nabla_\alpha Y^*_{\lambda\beta} - \frac{1}{3}[\nabla_\alpha,\nabla_\lambda]Y^{*\lambda}{}_\beta \tag{153}$$

$$= \frac{2}{3}\nabla^\lambda\left(g_{\alpha[\lambda}\xi_{\beta]}\right) - \frac{1}{3}g^{\lambda\rho}[\nabla_\alpha,\nabla_\lambda]Y^*_{\rho\beta} \tag{154}$$

$$= \frac{1}{3}\nabla_\alpha \xi_\beta - \frac{1}{3}g_{\alpha\beta}\nabla^\lambda \xi_\lambda$$
$$+ \frac{1}{3}\left(R^\lambda{}_\alpha Y^*_{\lambda\beta} + R^\lambda{}_{\beta\alpha}{}^\rho Y^*_{\rho\lambda}\right). \tag{155}$$

This gives rise to the Killing-Yano equation *first integrability condition*:

$$\boxed{\nabla_\alpha \xi_\beta = \frac{1}{2}\left(R^\lambda{}_\alpha Y^*_{\lambda\beta} + R^\lambda{}_{\alpha\beta}{}^\rho Y^*_{\lambda\rho}\right)} \tag{156}$$

or, equivalently, using Lemma 3:

$$\boxed{\nabla_\alpha \xi_\beta = \frac{1}{2}R^\lambda{}_\alpha Y^*_{\lambda\beta} - \frac{1}{4}R_{\alpha\beta\mu\nu}Y^{*\mu\nu}.} \tag{157}$$

Symmetrizing this equation gives the reduced form

$$\boxed{\nabla_{(\alpha}\xi_{\beta)} = \frac{1}{2}Y^*_{\lambda(\alpha}R^\lambda{}_{\beta)}.} \tag{158}$$

In particular, for Ricci-flat spacetimes, $\xi^\mu$ is a Killing vector.

A second integrability condition can be written as follows. Taking the derivative of the defining equation $\nabla_{(\alpha}Y_{\beta)\gamma} = 0$ yields

$$\nabla_\alpha \nabla_\beta Y_{\gamma\delta} + \nabla_\alpha \nabla_\gamma Y_{\beta\delta} = 0. \tag{159}$$

Out of this equation, we can write the three (equivalent) identities

$$\nabla_\alpha \nabla_\beta Y_{\gamma\delta} + \nabla_\gamma \nabla_\alpha Y_{\beta\delta} + [\nabla_\alpha,\nabla_\gamma]Y_{\beta\delta} = 0, \tag{160}$$

$$\nabla_\gamma \nabla_\alpha Y_{\beta\delta} + \nabla_\beta \nabla_\gamma Y_{\alpha\delta} + [\nabla_\gamma,\nabla_\beta]Y_{\alpha\delta} = 0, \tag{161}$$

$$\nabla_\beta \nabla_\gamma Y_{\alpha\delta} + \nabla_\alpha \nabla_\beta Y_{\gamma\delta} + [\nabla_\beta,\nabla_\alpha]Y_{\gamma\delta} = 0. \tag{162}$$

Summing the first and the third and subtracting the second leads to

$$2\nabla_\alpha\nabla_\beta Y_{\gamma\delta} = [\nabla_\gamma,\nabla_\alpha]Y_{\beta\delta} + [\nabla_\gamma,\nabla_\beta]Y_{\alpha\delta} + [\nabla_\alpha,\nabla_\beta]Y_{\gamma\delta} \tag{163}$$

$$= R^\lambda{}_{\delta\beta\gamma}Y_{\alpha\lambda} + R^\lambda{}_{\delta\alpha\gamma}Y_{\beta\lambda} + R^\lambda{}_{\delta\beta\alpha}Y_{\gamma\lambda} - \left(R^\lambda{}_{\beta\gamma\alpha} + R^\lambda{}_{\alpha\gamma\beta} + R^\lambda{}_{\gamma\alpha\beta}\right)Y_{\lambda\delta} \tag{164}$$

$$= 2R^\lambda{}_{\alpha\beta\gamma}Y_{\lambda\delta} + R^\lambda{}_{\delta\beta\gamma}Y_{\alpha\lambda} + R^\lambda{}_{\delta\alpha\gamma}Y_{\beta\lambda} + R^\lambda{}_{\delta\beta\alpha}Y_{\gamma\lambda}. \tag{165}$$

This gives rise to the *second integrability condition*

$$\boxed{\nabla_\alpha\nabla_\beta Y_{\gamma\delta} = R^\lambda{}_{\alpha\beta\gamma}Y_{\lambda\delta} + \frac{1}{2}\left(R^\lambda{}_{\delta\beta\gamma}Y_{\alpha\lambda} + R^\lambda{}_{\delta\alpha\gamma}Y_{\beta\lambda} + R^\lambda{}_{\delta\beta\alpha}Y_{\gamma\lambda}\right).} \tag{166}$$

A reduced form can be obtained by contracting the equation above with $g^{\gamma\delta}$. We obtain

$$0 = R^\lambda{}_{\alpha\beta}{}^\rho Y_{\lambda\rho} + \frac{1}{2}\left(R^\lambda{}_\beta Y_{\alpha\lambda} + R^\lambda{}_\alpha Y_{\beta\lambda} + R^{\lambda\rho}{}_{\beta\alpha}Y_{\rho\lambda}\right). \tag{167}$$

Using Lemma 3 shows that the first and the fourth terms of the right-hand side of this relation cancel. We obtain the *reduced form of the second integrability condition*

$$\boxed{Y_{\lambda(\alpha}R^\lambda{}_{\beta)} = 0.} \tag{168}$$

We can also symmetrize the indices $\gamma\delta$ in Eq. (166) to obtain the *symmetrized second integrability condition*

$$\boxed{R^\lambda{}_{\alpha\beta(\gamma}Y_{\delta)\lambda} - \frac{1}{2}Y_{\lambda(\gamma}R^\lambda{}_{\delta)\alpha\beta} + R^\lambda{}_{(\gamma\delta)(\alpha}Y_{\beta)\lambda} = 0.} \tag{169}$$

### B. The central identity

Our so-called central identity will consists into a clever rewriting of the expression $K_{\lambda(\beta}R^{*\lambda}{}_{\gamma\delta)\alpha}$. Its reduced version first introduced by Rüdiger [23] is a rewriting of the contracted expression $K_{\lambda\rho}R^{*\lambda}{}_{\alpha\beta}{}^\rho$.

#### 1. The KY scalar

Let us define the scalar quantity

$$\mathcal{Z} \triangleq \frac{1}{4}Y^*_{\alpha\beta}Y^{\alpha\beta}. \tag{170}$$

Its first covariant derivative takes the form

$$\nabla_\mu\mathcal{Z} = \frac{1}{2}\nabla_\mu Y^*_{\alpha\beta}Y^{\alpha\beta} \tag{171}$$

$$= \xi_{[\alpha}g_{\beta]\mu}Y^{\alpha\beta} \tag{172}$$

$$= \xi_\alpha Y^\alpha{}_\mu. \tag{173}$$

Its second covariant derivative can be expressed as (notice that $\nabla_\alpha\nabla_\beta\mathcal{Z} = \nabla_\beta\nabla_\alpha\mathcal{Z}$)

$$\nabla_\mu\nabla_\nu\mathcal{Z} = \nabla_\mu\xi_\alpha Y^\alpha{}_\nu + \xi^\alpha\varepsilon_{\mu\alpha\nu\rho}\xi^\rho \tag{174}$$

$$= \nabla_\mu\xi_\alpha Y^\alpha{}_\nu = \nabla_\nu\xi_\alpha Y^\alpha{}_\mu, \tag{175}$$

where the last equality follows from symmetry of the right-hand side.

The following identity is also useful:

$$\boxed{Y_{\alpha[\beta}Y_{\gamma]\delta} = -\frac{1}{2}Y_{\beta\gamma}Y_{\alpha\delta} - \frac{1}{2}\mathcal{Z}\,\varepsilon_{\alpha\beta\gamma\delta}.} \tag{176}$$

The proof consists into noticing that the combination $Y_{\alpha\beta}Y_{\gamma\delta} - Y_{\alpha\gamma}Y_{\beta\delta} + Y_{\beta\gamma}Y_{\alpha\delta}$ is totally antisymmetric in all its indices. It must consequently be proportional to the Levi-Civita tensor:

$$Y_{\alpha\beta}Y_{\gamma\delta} - Y_{\alpha\gamma}Y_{\beta\delta} + Y_{\beta\gamma}Y_{\alpha\delta} = \mathcal{A}\,\varepsilon_{\alpha\beta\gamma\delta}, \tag{177}$$

where the constant $\mathcal{A}$ remains to be determined. This is achieved by contracting the equation above with $\varepsilon^{\alpha\beta\gamma\delta}$, yielding

$$3\varepsilon^{\alpha\beta\gamma\delta}Y_{\alpha\beta}Y_{\gamma\delta} = -4!\,\mathcal{A}. \tag{178}$$

Using the definition of $\mathcal{Z}$ leads to

$$\mathcal{A} = -\mathcal{Z}, \tag{179}$$

which gives the desired result.

#### 2. Derivation of the central identity

The trick for deriving the central identity is to define the tensor

$$\mathcal{T}_{\mu\nu\rho\sigma} = \varepsilon_{\mu\alpha\beta\sigma}\nabla^\alpha\nabla^\beta Y_{\nu\lambda}Y^\lambda{}_\rho, \tag{180}$$

to perform two different rewritings of this expression and finally to equate them. Notice that we recover the tensor used in Rüdiger's proof [23] by contracting the last two indices of Eq. (180).

First, applying the Ricci identity to Eq. (180) and making use of Eq. (176) yields

$$\mathcal{T}_{\mu\nu\rho\sigma} = \frac{1}{2}\varepsilon_{\mu\alpha\beta\sigma}\left[\nabla^\alpha, \nabla^\beta\right]Y_{\nu\lambda}Y^\lambda{}_\rho = -\left(R^{*\kappa}{}_{\nu\mu\sigma}K_{\kappa\rho} + R^{*\kappa\lambda}{}_{\mu\sigma}Y_{\nu[\kappa}Y_{\lambda]\rho}\right) \tag{181}$$

$$= R^{*\kappa}{}_{\nu\sigma\mu}K_{\kappa\rho} + \frac{1}{2}R^{*\kappa\lambda}{}_{\mu\sigma}\left(Y_{\kappa\lambda}Y_{\nu\rho} + \mathcal{Z}\varepsilon_{\nu\kappa\lambda\rho}\right). \tag{182}$$

Second, using Eqs. (138), (141) and Lemma 1, we rewrite $\mathcal{T}_{\mu\nu\rho\sigma}$ as follows:

$$\mathcal{T}^\mu{}_{\nu\rho}{}^\sigma = -\frac{1}{2}\varepsilon^{\mu\alpha\beta\sigma}\varepsilon_{\nu\lambda\gamma\delta}\nabla_\alpha\nabla_\beta Y^{*\gamma\delta}Y^\lambda{}_\rho \tag{183}$$

$$= -12\delta_\nu^{[\sigma}Y^\mu{}_\rho\nabla_\alpha\nabla_\beta Y^{*\alpha\beta]} \tag{184}$$

$$= -\delta_\nu^\sigma\left(Y^\mu{}_\rho\nabla_\alpha\nabla_\beta Y^{*\alpha\beta} + Y^\alpha{}_\rho\nabla_\alpha\nabla_\beta Y^{*\beta\mu} + Y^\beta{}_\rho\nabla_\alpha\nabla_\beta Y^{*\mu\alpha}\right)$$
$$+ \delta_\nu^\mu\left(Y^\sigma{}_\rho\nabla_\alpha\nabla_\beta Y^{*\alpha\beta} + Y^\alpha{}_\rho\nabla_\alpha\nabla_\beta Y^{*\beta\sigma} + Y^\beta{}_\rho\nabla_\alpha\nabla_\beta Y^{*\sigma\alpha}\right)$$
$$- \delta_\nu^\alpha\left(Y^\sigma{}_\rho\nabla_\alpha\nabla_\beta Y^{*\mu\beta} + Y^\mu{}_\rho\nabla_\alpha\nabla_\beta Y^{*\beta\sigma} + Y^\beta{}_\rho\nabla_\alpha\nabla_\beta Y^{*\sigma\mu}\right)$$
$$+ \delta_\nu^\beta\left(Y^\sigma{}_\rho\nabla_\alpha\nabla_\beta Y^{*\mu\alpha} + Y^\mu{}_\rho\nabla_\alpha\nabla_\beta Y^{*\alpha\sigma} + Y^\alpha{}_\rho\nabla_\alpha\nabla_\beta Y^{*\sigma\mu}\right) \tag{185}$$

$$= \delta_\nu^\sigma\left(3Y^\alpha{}_\rho\nabla_\alpha\xi^\mu + 2Y^\beta{}_\rho\delta_\beta^{[\mu}\nabla_\alpha\xi^{\alpha]}\right) - \delta_\nu^\mu\left(3Y^\alpha{}_\rho\nabla_\alpha\xi^\sigma + 2Y^\beta{}_\rho\delta_\beta^{[\sigma}\nabla_\alpha\xi^{\alpha]}\right)$$
$$- 3Y^\sigma{}_\rho\nabla_\nu\xi^\mu + 3Y^\mu{}_\rho\nabla_\nu\xi^\sigma + 2Y^\beta{}_\rho\delta_\beta^{[\sigma}\nabla_\nu\xi^{\mu]} - 2\left(Y^\sigma{}_\rho\delta_\nu^{[\mu}\nabla_\alpha\xi^{\alpha]} + Y^\mu{}_\rho\delta_\nu^{[\alpha}\nabla_\alpha\xi^{\sigma]} + Y^\alpha{}_\rho\delta_\nu^{[\sigma}\nabla_\alpha\xi^{\mu]}\right) \tag{186}$$

$$= \delta_\nu^\sigma Y^\lambda{}_\rho\nabla_\lambda\xi^\mu - \delta_\nu^\mu Y^\lambda{}_\rho\nabla_\lambda\xi^\sigma + Y^\mu{}_\rho\nabla_\nu\xi^\sigma - Y^\sigma{}_\rho\nabla_\nu\xi^\mu. \tag{187}$$

Equating the two expressions of $\mathcal{T}_{\mu\nu\rho\sigma}$ obtained leads to

$$R^{*\lambda}{}_{\nu\mu\sigma}K_{\rho\lambda} = \frac{1}{2}R^{*\kappa\lambda}{}_{\mu\sigma}\left(Y_{\kappa\lambda}Y_{\nu\rho} + \mathcal{Z}\varepsilon_{\nu\kappa\lambda\rho}\right) + g_{\mu\nu}Y^\lambda{}_\rho\nabla_\lambda\xi_\sigma - g_{\nu\sigma}Y^\lambda{}_\rho\nabla_\lambda\xi_\mu + Y_{\sigma\rho}\nabla_\nu\xi_\mu - Y_{\mu\rho}\nabla_\nu\xi_\sigma. \tag{188}$$

This is the cornerstone equation for deriving both the central identity and its reduced form.

*a. Non-reduced form.* Fully symmetrizing Eq. (188) in $(\mu\nu\rho)$ leads to

$$K_{\lambda(\beta}R^{*\lambda}{}_{\gamma\delta)\alpha} = Y^\lambda{}_{(\beta}g_{\gamma\delta)}\xi_{\alpha;\lambda} - g_{\alpha(\beta}Y^\lambda{}_\gamma\xi_{\delta);\lambda} + Y_{\alpha(\beta}\xi_{\gamma;\delta)}. \tag{189}$$

We make use of the reduced integrability condition (158) to write the last term as

$$Y_{\alpha(\beta}\xi_{\gamma;\delta)} = \frac{1}{2}Y_{\alpha(\beta|}Y^*_{\lambda|\gamma}R^\lambda{}_{\delta)} \tag{190}$$

$$= \frac{1}{2}Y_{\alpha(\beta}Y^{*\lambda}{}_\gamma G_{\delta)\lambda} + \frac{R}{4}Y_{\alpha(\beta}Y^*_{\gamma\delta)} \tag{191}$$

$$= \frac{1}{2}Y_{\alpha(\beta}Y^{*\lambda}{}_\gamma G_{\delta)\lambda} \tag{192}$$

where $G_{\alpha\beta} \triangleq R_{\alpha\beta} - \frac{R}{2}g_{\alpha\beta}$ is the Einstein tensor. This gives rise to the central identity:

$$\boxed{K_{\lambda(\beta}R^{*\lambda}{}_{\gamma\delta)\alpha} = Y^\lambda{}_{(\beta}g_{\gamma\delta)}\xi_{\alpha;\lambda} - g_{\alpha(\beta}Y^\lambda{}_\gamma\xi_{\delta);\lambda} + \frac{1}{2}Y_{\alpha(\beta}Y^{*\lambda}{}_\gamma G_{\delta)\lambda}.} \tag{193}$$

*b. Reduced form.* Rüdiger's reduced form of the central identity can be derived by contracting Eq. (188) with $g^{\rho\sigma}$ and using Eqs. (158) and (175):

$$R^{*\lambda}{}_{\nu\mu}{}^\rho K_{\lambda\rho} = \frac{1}{2}R^{*\kappa\lambda}{}_\mu{}^\rho\left(Y_{\kappa\lambda}Y_{\nu\rho} + \mathcal{Z}\varepsilon_{\nu\kappa\lambda\rho}\right) + g_{\mu\nu}Y^{\lambda\rho}\nabla_\lambda\xi_\rho - Y^\lambda{}_\nu\nabla_\lambda\xi_\mu - Y_{\mu\rho}\nabla_\nu\xi^\rho \tag{194}$$

$$= \frac{1}{2}R^{*\kappa\lambda}{}_\mu{}^\rho\left(Y_{\kappa\lambda}Y_{\nu\rho} + \mathcal{Z}\varepsilon_{\nu\kappa\lambda\rho}\right) - g_{\mu\nu}Y^{\lambda\rho}\nabla_\rho\xi_\lambda + Y^\lambda{}_\nu\nabla_\mu\xi_\lambda - Y^\lambda{}_\nu Y^*_{\rho(\lambda}R^\rho{}_{\mu)} - Y_{\mu\rho}\nabla_\nu\xi^\rho \tag{195}$$

$$= \frac{1}{2}R^{*\kappa\lambda}{}_\mu{}^\rho\left(Y_{\kappa\lambda}Y_{\nu\rho} + \mathcal{Z}\varepsilon_{\nu\kappa\lambda\rho}\right) + 2\nabla_\mu\nabla_\nu\mathcal{Z} - g_{\mu\nu}\Delta\mathcal{Z} - Y^\lambda{}_\nu Y^*_{\rho(\lambda}G^\rho{}_{\mu)}. \tag{196}$$

The first term of the above equation can be simplified by noticing that, on the one hand, making use of Lemma 3,

$$\frac{1}{2}R^{*\kappa\lambda}{}_\mu{}^\rho Y_{\kappa\lambda}Y_{\nu\rho} = \frac{1}{4}\varepsilon_\mu{}^{\alpha\beta\lambda}R^{\sigma\rho}{}_{\alpha\beta}Y_{\rho\sigma}Y_{\lambda\nu} = -\frac{1}{4}\varepsilon_\mu{}^{\alpha\beta\lambda}R^{\sigma\rho}{}_{\alpha\beta}Y^{**}_{\rho\sigma}Y_{\lambda\nu} \tag{197}$$

$$= -\frac{1}{8}\varepsilon_{\mu\alpha\beta\lambda}\varepsilon^{\rho\sigma\gamma\delta}R^{\alpha\beta}{}_{\sigma\rho}Y^*_{\gamma\delta}Y^\lambda{}_\nu = 3R^{\alpha\beta}{}_{[\alpha\mu}Y^*_{\beta\lambda]}Y^\lambda{}_\nu \tag{198}$$

$$= \frac{1}{4}Y^\lambda{}_\nu\left(4R^\beta{}_\mu Y^*_{\beta\lambda} + 4R^\beta{}_\lambda Y^*_{\mu\beta} - 2R^{\alpha\beta}{}_{\mu\lambda}Y^*_{\alpha\beta} - 2RY^*_{\mu\lambda}\right) \tag{199}$$

$$= \frac{1}{4}Y^\lambda{}_\nu\left[4\left(R^\alpha{}_{\mu\lambda}{}^\beta Y^*_{\alpha\beta} + R^\beta{}_\mu Y^*_{\beta\lambda}\right) + 4R^\beta{}_\lambda Y^*_{\mu\beta} - 2RY^*_{\mu\lambda}\right] \tag{200}$$

$$\overset{(157)}{=} 2Y^\lambda{}_\nu\nabla_\mu\xi_\lambda + R^\beta{}_\lambda Y^\lambda{}_\nu Y^*_{\mu\beta} - \frac{1}{2}RY^\lambda{}_\nu Y^*_{\mu\lambda} \tag{201}$$

$$= 2\nabla_\mu\nabla_\nu\mathcal{Z} + G^\beta{}_\lambda Y^\lambda{}_\nu Y^*_{\mu\beta}. \tag{202}$$

On the other hand, the term $\frac{1}{2}R^{*\kappa\lambda}{}_\mu{}^\rho\mathcal{Z}\varepsilon_{\nu\kappa\lambda\rho}$ can be reduced thanks to the identity

$$\frac{1}{4}\mathcal{Z}\varepsilon^{\mu\alpha\beta\lambda}\varepsilon_{\lambda\rho\sigma\nu}R^{\sigma\rho}{}_{\alpha\beta} = \frac{3}{2}\mathcal{Z}\delta^\mu_{[\rho}R^{\sigma\rho}{}_{\sigma\nu]} = \frac{1}{2}\mathcal{Z}\left(2R^\mu{}_\nu - \delta^\mu_\nu R\right) = \mathcal{Z}G^\mu{}_\nu. \tag{203}$$

Putting all pieces together, we obtain the reduced central identity

$$\boxed{R^{*\lambda}{}_{\nu\mu}{}^\rho K_{\lambda\rho} = 4\nabla_\mu\nabla_\nu\mathcal{Z} - g_{\mu\nu}\Delta\mathcal{Z} - Y^{*\lambda}{}_{(\rho}G_{\mu)\lambda}Y^\rho{}_\nu + G^\beta{}_\lambda Y^\lambda{}_\nu Y^*_{\mu\beta} + \mathcal{Z}G_{\mu\nu}.} \tag{204}$$

This is the generalization of Rüdiger's reduced central identity [23] to non Ricci-flat spacetimes.

### 3. Central identity in Ricci-flat spacetimes.

Let us now particularize our analysis to vacuum spacetimes, i.e. Ricci-flat spacetimes $R_{\alpha\beta} = G_{\alpha\beta} = 0$. This includes in particular the astrophysically relevant Kerr spacetime.

Using the reduced integrability condition (158), the central identity (193) becomes

$$K_{\lambda(\beta}R^{*\lambda}{}_{\gamma\delta)\alpha} = Y^\lambda{}_{(\beta}g_{\gamma\delta)}\xi_{\alpha;\lambda} - g_{\alpha(\beta}Y^\lambda{}_\gamma\xi_{\delta);\lambda} \tag{205}$$

$$= -Y^\lambda{}_{(\beta}g_{\gamma\delta)}\xi_{\lambda;\alpha} + g_{\alpha(\beta}Y^\lambda{}_{\gamma|}\xi_{\lambda;|\delta)}. \tag{206}$$

Making use of Eq. (175), it takes the final form

$$\boxed{K_{\lambda(\beta}R^{*\lambda}{}_{\gamma\delta)\alpha} = -\nabla_\alpha\nabla_{(\beta}\mathcal{Z}g_{\gamma\delta)} + g_{\alpha(\beta}\nabla_\gamma\nabla_{\delta)}\mathcal{Z},} \tag{207}$$

which does not appear in Rüdiger [23]. The reduced central identity (204) becomes

$$\boxed{R^{*\lambda}{}_{\mu\nu}{}^\rho K_{\lambda\rho} = 4\nabla_\mu\nabla_\nu\mathcal{Z} - g_{\mu\nu}\Delta\mathcal{Z},} \tag{208}$$

as obtained by Rüdiger [23].

## VI. SOLUTIONS TO THE $\mathcal{O}(\mathcal{S})$ CONSTRAINT IN RICCI-FLAT SPACETIMES

In this Section, one will gather the results obtained in the two previous sections of this paper. Our aim will be to solve the constraint (134) in Ricci-flat spacetimes admitting a KY tensor. The Ricci-flatness assumption will enable us to make use of the simple expressions provided by the central identity (207) and its reduced form (208).

### A. Simplification of the constraint

Plugging Eq. (208) into Eq. (134) leads – after a few easy manipulations – to the constraint

$$^*X_{\alpha(\beta\gamma;\delta)} + \nabla_{(\beta|}\left[X_\alpha - \frac{4}{3}\nabla_\alpha\mathcal{Z}\right]g_{|\gamma\delta)} - g_{\alpha(\beta}\nabla_\delta\left[X_{\gamma)} - \frac{4}{3}\nabla_{\gamma)}\mathcal{Z}\right] = 0. \tag{209}$$

It is straightforward to see that it admits the following non-trivial solution

$$X_{\alpha\beta\gamma} = 0, \qquad X_\alpha = \frac{4}{3}\nabla_\alpha \mathcal{Z}. \tag{210}$$

As we will see later, this solution will lead to Rüdiger's invariant (68). In what follows, we will seek a more general solution to Eq. (209), which does not assume $X_{\alpha\beta\gamma} = 0$. The first step is to simplify more the constraint (209). We begin by rewriting it in the much simpler form

$$\left[{}^*X_{\alpha(\beta\gamma} + Y_\alpha g_{(\beta\gamma} - g_{\alpha(\beta} Y_{\gamma)}\right]_{;\delta)} = 0 \tag{211}$$

by introducing the shifted variable

$$Y_\alpha \triangleq X_\alpha - \frac{4}{3}\nabla_\alpha \mathcal{Z}. \tag{212}$$

We subsequently rewrite Eq. (211) in order to remove the dual operator from the tensor **X**. Let us define the Hodge dual $\tilde{Y}_{\alpha\beta\gamma}$ of $Y_\alpha$:

$$\tilde{Y}_{\alpha\beta\gamma} \triangleq \varepsilon_{\mu\alpha\beta\gamma} Y^\mu \qquad \Leftrightarrow \qquad Y_\alpha \triangleq -\frac{1}{6}\varepsilon_{\alpha\mu\nu\rho}\tilde{Y}^{\mu\nu\rho}. \tag{213}$$

Contracting Eq. (211) with $\varepsilon^{\alpha\mu\nu\rho}$ and using the usual properties of the Levi-Civita tensor, we get

$$-3\delta^{[\mu}_{(\beta}X^{\nu\rho]}{}_{\gamma;\delta)} + \tilde{Y}^{\mu\nu\rho}{}_{;(\beta}g_{\gamma\delta)} - \varepsilon_{(\beta}{}^{\mu\nu\rho}Y_{\gamma;\delta)} = 0. \tag{214}$$

Now, using the fact that

$$\varepsilon^{\beta\mu\nu\rho}Y_{\gamma;\delta} = 4\delta^{[\beta}_\gamma \tilde{Y}^{\mu\nu\rho]}{}_{;\delta}, \tag{215}$$

the last term of the previous equation reads as

$$-\varepsilon_{(\beta}{}^{\mu\nu\rho}Y_{\gamma;\delta)} = -g_{(\beta\gamma}\tilde{Y}^{\mu\nu\rho}{}_{;\delta)} + 3\delta^{[\mu}_{(\gamma}\tilde{Y}^{\nu\rho]}{}_{\beta;\delta)}. \tag{216}$$

Putting all pieces together, Eq. (209) becomes equivalent to

$$\delta^{[\mu}_{(\beta}\left(X^{\nu\rho]}{}_\gamma - \tilde{Y}^{\nu\rho]}{}_\gamma\right)_{;\delta)} = 0. \tag{217}$$

It follows from Eq. (99) and from the definition of $L_{\alpha\beta\gamma} = L_{[\alpha\beta]\gamma}$ that $X_{\alpha\beta\gamma}$ is antisymmetric on its two first indices, $X_{\alpha\beta\gamma} = X_{[\alpha\beta]\gamma}$. Let us decompose $X_{\alpha\beta\gamma}$ into its traces and trace-free parts:

$$X_{\alpha\beta\gamma} = A_\alpha g_{\beta\gamma} + B_\beta g_{\alpha\gamma} + C_\gamma g_{\alpha\beta} + D^{\text{tf}}_{\alpha\beta\gamma}. \tag{218}$$

Note that the constraint $X_{[\alpha\beta\gamma]} = 0$ reduces to $D^{\text{tf}}_{[\alpha\beta\gamma]} = 0$. Moreover, since $X_{\alpha\beta\gamma} = X_{[\alpha\beta]\gamma}$, it implies that one can set $C_\gamma = 0$. Plugging the above decomposition into Eq.

(217) and using the identity $\delta^{[\mu}_{(\alpha}\delta^{\nu]}_{\beta)} = 0$, the constraint (217) becomes

$$\delta^{[\mu}_{(\beta}\left(D^{\text{tf}\,\nu\rho]}{}_\gamma - \tilde{Y}^{\nu\rho]}{}_\gamma\right)_{;\delta)} = 0. \tag{219}$$

This implies that the 1-forms $A_\alpha$ and $B_\beta$ determining the trace part of $X_{\alpha\beta\gamma}$ are left unconstrained. However, these trace parts produce terms into the conserved quantity (69) containing a $p_\mu S^{\mu\nu}$ factor, which vanish due to the spin supplementary condition (13). All in all, we can, without loss of generality, set $A_\alpha = B_\beta = 0 = C_\gamma$, *i.e.* consider that $X_{\alpha\beta\gamma}$ reduces to its traceless part. The constraint equation can be written in the short form

$$\delta^{[\mu}_{(\beta}W^{\nu\rho]}{}_{\gamma;\delta)} = 0, \tag{220}$$

where we have defined $W_{\alpha\beta\gamma} \triangleq D^{\text{tf}}_{\alpha\beta\gamma} - \tilde{Y}_{\alpha\beta\gamma}$, which is by definition traceless and antisymmetric in its two first indices. Contracting the constraint (220) with $-\frac{1}{2}\varepsilon_{\mu\nu\rho\alpha}$ leads to the equivalent condition

$$^*W_{\alpha(\beta\gamma;\delta)} = 0, \tag{221}$$

where

$$^*W_{\alpha\beta\gamma} = {}^*D^{\text{tf}}_{\alpha\beta\gamma} - 2g_{\gamma[\alpha}Y_{\beta]}. \tag{222}$$

Given a solution to the simplified constraint (221), the quasi-invariant $K_{\mu\nu}p^\mu p^\nu + L_{\alpha\beta\gamma}S^{\alpha\beta}p^\gamma$ can be constructed using Eqs. (99), (101) and (218) which leads to

$$L_{\alpha\beta\gamma} = D^{\text{tf}}_{\alpha\beta\gamma} + \frac{1}{3}\lambda_{\alpha\beta\gamma} + \varepsilon_{\alpha\beta\gamma\delta}(Y^\delta + \frac{4}{3}\nabla^\delta Z). \tag{223}$$

### B. The Rüdiger quasi-invariant $\mathcal{Q}_R$

Let us first recover Rüdiger's quasi-invariant (68). In terms of our new variables, Rüdiger's solution (210) is simply

$$D^{\text{tf}}_{\alpha\beta\gamma} = 0, \qquad Y_\alpha = 0. \tag{224}$$

Substituting in (223), it leads to the quasi-invariant

$$\mathcal{Q}_R = K_{\mu\nu}p^\mu p^\nu + \frac{1}{3}\lambda_{\mu\nu\rho}S^{\mu\nu}p^\rho + \frac{4}{3}\varepsilon_{\mu\nu\rho\sigma}S^{\mu\nu}p^\rho\nabla_\sigma \mathcal{Z}. \tag{225}$$

Let us work on each term of the right-hand side separately:

- We recall the definition $L_\alpha \triangleq Y_{\alpha\lambda}p^\lambda$. The definition (136) leads to

$$K_{\mu\nu}p^\mu p^\nu = -L_\alpha L^\alpha. \tag{226}$$

- Using the definitions (100) and (136) and the properties (137) and (173) we have

$$\lambda_{\mu\nu\rho}S^{\mu\nu}p^\rho = 2\nabla_\mu K_{\nu\rho}S^{\mu\nu}p^\rho = -2\left(\varepsilon_{\mu\nu\lambda\sigma}\varepsilon^{\mu\nu\alpha\beta}Y^\lambda{}_\rho + \varepsilon_{\mu\lambda\rho\sigma}\varepsilon^{\mu\nu\alpha\beta}Y_\nu{}^\lambda\right)\xi^\sigma\hat{p}_\alpha S_\beta p^\rho \tag{227}$$

$$= 2\left(4Y^\lambda{}_\rho\xi^\sigma\hat{p}_{[\lambda}S_{\sigma]}p^\rho + 6Y_{[\lambda}{}^\lambda\hat{p}_\rho S_{\sigma]}\xi^\sigma p^\rho\right) \tag{228}$$

$$= 2\left(-3Y^\lambda{}_\rho p^\rho\xi^\sigma\hat{p}_\sigma S_\lambda - Y_\sigma{}^\lambda S_\lambda\xi^\sigma\hat{p}_\rho p^\rho + \underbrace{Y_\sigma{}^\lambda\hat{p}_\lambda\xi^\sigma S_\rho p^\rho}_{=0}\right) \tag{229}$$

$$= 2\mu\xi^\sigma Y_\sigma{}^\lambda S_\lambda - 6\mu^{-1}L_\alpha S^\alpha\xi_\beta p^\beta = 2\mu S^\alpha\partial_\alpha\mathcal{Z} - 6\mu^{-1}L_\alpha S^\alpha\xi_\beta p^\beta. \tag{230}$$

Using (76), the factor $L_\alpha S^\alpha$ can be written in a much more enlightening way:

$$L_\alpha S^\alpha = Y_{\alpha\lambda}p^\lambda S^\alpha = Y^*_{\alpha\lambda}p^{[\alpha}S^{\lambda]*} = -\frac{\mu}{2}Y^*_{\alpha\lambda}S^{\alpha\lambda}$$
$$= -\frac{\mu}{2}\mathcal{Q}_Y, \tag{231}$$

where $\mathcal{Q}_Y$ is Rüdiger's linear invariant (66), which is also conserved up to linear order in the spin.

- Using the definition of the spin vector (22) together with the spin supplementary condition allows to write

$$\varepsilon_{\mu\nu\rho\sigma}S^{\mu\nu}p^\rho\nabla^\sigma\mathcal{Z} = 4\delta^{[\alpha}_\rho\delta^{\beta]}_\sigma\nabla^\sigma\mathcal{Z}\hat{p}_\alpha S_\beta p^\rho$$
$$= 4\hat{p}_{[\rho}S_{\sigma]}\nabla^\sigma\mathcal{Z}p^\rho$$
$$= 2S_\sigma\nabla^\sigma\mathcal{Z}\hat{p}_\rho p^\rho = -2\mu S^\alpha\partial_\alpha\mathcal{Z}. \tag{232}$$

Putting all the pieces together, we get the following form for $Q_R$,

$$\mathcal{Q}_R = -L_\alpha L^\alpha - 2\mu S^\alpha\partial_\alpha\mathcal{Z} + \mathcal{Q}_Y\,\xi_\beta p^\beta. \tag{233}$$

which is identically the quadratic Rüdiger invariant (68). For Ricci-flat spacetimes, $\xi^\mu$ is a Killing vector and $\xi_\alpha p^\alpha$ can be upgraded to an invariant as (65) with a $O(S^1)$ correction. This implies that the last term in (233) is trivial since it is a product of quasi-invariant at linear order in $S$.

### C. Trivial solutions to the algebraic constraints

Let us now discuss more general solutions to the simplified algebraic constraints (221). By linearity, we can substract the Rüdiger quasi-invariant (225) from the definition of quasi-invariants (69) where $L_{\alpha\beta\gamma}$ is given in Eq. (223). Given a solution to the simplified algebraic constraints (221), one therefore obtains a quasi-invariant of the form

$$\mathcal{Q}(D^{\text{tf}}_{\alpha\beta\gamma},Y_\alpha) = \left(D^{\text{tf}}_{\alpha\beta\gamma} + \varepsilon_{\alpha\beta\gamma\lambda}Y^\lambda\right)S^{\alpha\beta}p^\gamma \tag{234}$$

$$= 2\left({}^*D^{\text{tf}}_{\alpha\beta\gamma} + {}^*\varepsilon_{\alpha\beta\gamma\lambda}Y^\lambda\right)S^\alpha\hat{p}^\beta p^\gamma \tag{235}$$

$$= 2\mu\,{}^*D^{\text{tf}}_{\alpha(\beta\gamma)}S^\alpha\hat{p}^\beta\hat{p}^\gamma - 2\mu S_\alpha Y^\alpha \tag{236}$$

after using Eq. (42). Such quasi-invariant is homogeneous in $S$.

Looking at (221), it is appealing to first attempt to generate a quasi-invariant through solving the stronger algebraic constraint

$${}^*W_{\alpha(\beta\gamma)} = 0. \tag{237}$$

However, this procedure only leads to identically vanishing quasi-invariants. Indeed, by virtue of Eq. (222), one then has

$${}^*D^{\text{tf}}_{\alpha(\beta\gamma)} = g_{\alpha(\gamma}Y_{\beta)} - g_{\beta\gamma}Y_\alpha. \tag{238}$$

This yields

$$\mathcal{Q}(D^{\text{tf}}_{\alpha\beta\gamma},Y_\alpha) = -2\mu S_\alpha Y^\alpha\hat{p}_\beta\hat{p}^\beta - 2\mu S_\alpha Y^\alpha = 0. \tag{239}$$

This implies that new non-trivial quasi-invariants can only be generated by making a non-trivial use of the symmetrized covariant derivative $_{;\delta)}$ in Eq. (221).

### D. Invariants homogeneously linear in $\mathcal{S}$

Non-trivial invariants of the form (236) can only be generated by finding a solution to the differential constraint (221). As we will see in this section, this problem and the form of the generated invariant can be recast in a very simple form. The first step is to express the invariant (236) as a function of ${}^*W_{\alpha\beta\gamma}$ only. Contracting Eq. (222) with $g^{\beta\gamma}$ yields

$$\boxed{Y_\alpha = \frac{1}{3}\,{}^*W_{\alpha\lambda}{}^\lambda.} \tag{240}$$

Rearranging Eq. (222) and making use of Eq. (240), we get

$$\boxed{{}^*D^{\text{tf}}_{\alpha\beta\gamma} = {}^*W_{\alpha\beta\gamma} + \frac{2}{3}g_{\gamma[\alpha}\,{}^*W_{\beta]\lambda}{}^\lambda.} \tag{241}$$

Plugging these two expressions into Eq. (236) and using the orthogonality condition $\hat{p}_\alpha S^\alpha = 0$ leads to the simple expression

$$\boxed{\mathcal{Q}(W_{\alpha\beta\gamma}) = 2\mu\,{}^*W_{\alpha(\beta\gamma)}S^\alpha\hat{p}^\beta\hat{p}^\gamma.} \tag{242}$$

Now, because the dualization is an invertible operation, the giving of $^{*}W_{\alpha(\beta\gamma)}$ is equivalent to the giving of the symmetrized part in its two last indices of a rank-3 tensor $N_{\alpha\beta\gamma}$ antisymmetric in its two first indices, such that

$$N_{\alpha\beta\gamma} \equiv {}^{*}W_{\alpha\beta\gamma}, \tag{243}$$

which obeys

$$N_{\alpha(\beta\gamma;\delta)} = 0. \tag{244}$$

Note that one *cannot* impose that $N_{\alpha\beta\gamma}$ is also symmetric in $\beta\gamma$ otherwise it would vanish because one would have $N_{\alpha\beta\gamma} = -N_{\beta\alpha\gamma} = -N_{\beta\gamma\alpha} = N_{\gamma\beta\alpha} = N_{\gamma\alpha\beta} = -N_{\alpha\gamma\beta} = -N_{\alpha\beta\gamma}$.

The tensor $T_{\alpha\beta\gamma} \triangleq N_{\alpha(\beta\gamma)}$ obeys the cyclic identity

$$T_{\alpha\beta\gamma} + T_{\beta\gamma\alpha} + T_{\gamma\alpha\beta} = 0 \tag{245}$$

but since $T_{\alpha\beta\gamma}$ is not symmetric in its two first indices, it is not totally symmetric and the condition defining the mixed-symmetry tensor (244) is *distinct* from the condition defining a Killing tensor $K_{(\alpha\beta\gamma;\delta)} = 0$ where $K_{\alpha\beta\gamma}$ is totally symmetric $K_{(\alpha\beta\gamma)} = K_{\alpha\beta\gamma}$. In terms of representation of the permutation group, $T_{\alpha\beta\gamma}$ is a $\{2,1\}$ Young diagram.

Consequently, the following statement holds:

**Proposition 1.** *Let $(\mathcal{M}, g_{\mu\nu})$ be a (3+1)-dimensional Ricci-flat spacetime admitting a Killing-Yano tensor such that there exists on $\mathcal{M}$ a mixed-symmetry Killing tensor, i.e., a rank-3 tensor $T_{\alpha\beta\gamma}$ which is a $\{2,1\}$ Young tableau, i.e., built as $T_{\alpha\beta\gamma} = N_{\alpha(\beta\gamma)}$ such that $N_{\alpha\beta\gamma} = N_{[\alpha\beta]\gamma}$, satisfying the (differential) constraint*

$$T_{\alpha(\beta\gamma;\delta)} = 0. \tag{246}$$

*Then, the quantity*

$$\mathcal{N} \triangleq T_{\alpha\beta\gamma}S^{\alpha}\hat{p}^{\beta}\hat{p}^{\gamma} \tag{247}$$

*is a (homogeneously linear in $\mathcal{S}$) quasi-invariant for the linearized MPT equations on $\mathcal{M}$, i.e.*

$$\frac{d\mathcal{N}}{d\tau} = \mathcal{O}(\mathcal{S}^{2}). \tag{248}$$

### E. Trivial mixed-symmetry Killing tensors

If a mixed-symmetry Killing tensor $T_{\alpha\beta\gamma}$ is found, the only point to be addressed before claiming the existence of a new quasi-invariant is to check its non-triviality, *i.e.*, it should not be the product of two others quasi-invariants. We define a *trivial mixed-symmetry Killing tensor* $T_{\alpha\beta\gamma}$ as mixed-symmetry Killing tensor that generates a trivial quasi-invariant, *i.e.* an invariant which

can be written as the product of other quasi-invariants.[5] A *non-trivial mixed-symmetry Killing tensor* is a mixed-symmetry Killing tensor which is not trivial.

If the spacetime admits a Killing-Yano tensor and a Killing vector, we can construct a trivial mixed-symmetry Killing tensor of the form

$$T_{\alpha\beta\gamma} = Y_{\alpha(\beta}\xi_{\gamma)}. \tag{249}$$

built as $T_{\alpha\beta\gamma} = N^{(1)}_{\alpha(\beta\gamma)} = N^{(2)}_{\alpha(\beta\gamma)}$ from either

$$N^{(1)}_{\alpha\beta\gamma} = Y_{\alpha\beta}\xi_{\gamma}, \quad \text{or} \quad N^{(2)}_{\alpha\beta\gamma} = Y_{\alpha\gamma}\xi_{\beta} - Y_{\beta\gamma}\xi_{\alpha}. \tag{250}$$

It is straightforward from Eq. (135) and the Killing equation that they obey (246).

Now, for this $T_{\alpha\beta\gamma}$ the associated quasi-invariant is the following product of quasi-invariants:

$$Q = -\frac{1}{2}\mathcal{Q}_{Y}\,\xi^{\alpha}\hat{p}_{\alpha}, \tag{251}$$

where the linear Rüdiger quasi-invariant $\mathcal{Q}_{Y}$ is defined in Eq. (66) and obeys Eq. (231).

The question of existence of quasi-invariants beyond the ones found by Rüdiger therefore amounts to determine the existence of non-trivial mixed-symmetry Killing tensors.

In order to gain some intuition about mixed-symmetry Killing tensors, it is useful to derive the general such tensor in Minkowski spacetime. For a Riemann flat spacetime, the covariant derivative becomes a coordinate derivative in Minkowskian coordinates (in any dimension). We can therefore consider the index $\alpha$ in Eq. (246) as a parametric index and consider the list of 2-components objects $K_{(\alpha)\beta\gamma} = T_{\alpha\beta\gamma}$ symmetric under the exchange of $\beta\gamma$, parametrized by $\alpha$. The constraint (246) is then equivalent to the Killing tensor equation for each $K_{(\alpha)\beta\gamma}$, $\alpha$ fixed. Since all Killing tensors in Minkowski spacetime are direct products of Killing vectors, we can write $K_{(\alpha)\beta\gamma}$ as a sum of terms of the form $K_{(\alpha)\beta\gamma} = M_{(\alpha)(i)(j)}\xi^{(i)}_{\beta}\xi^{(j)}_{\gamma}$ where $(i), (j)$ label the independent Killing vectors. In order to respect the symmetries of a mixed-symmetry tensor we can write in particular the resulting $T_{\alpha\beta\gamma}$ as a linear combination of terms $X_{\alpha(\beta}\xi_{\gamma)}$ where $\xi^{\gamma}$ is a Killing vector. The symmetry properties of a mixed-symmetry tensor imply that $X_{\alpha\beta} = X_{[\alpha\beta]}$. The constraint (246) finally reduces to the condition that $X_{\alpha\beta}$ is a Killing-Yano tensor. We have therefore proven that any mixed-symmetry tensor in Minkowski spacetime takes the trivial form (249).

Now, Kerr spacetime admits a non-trivial Killing tensor and curvature and further analysis is required. We will address the existence of a mixed-symmetry tensor for Kerr spacetime in the following.

---

[5] It remains to be investigated if such a tensor necessarily takes the form of a sum of direct products of tensors such that Eq. (246) holds. We have not found any simple argument for proving this assertion.

## VII.   LINEAR INVARIANTS FOR KERR SPACETIME

### A.   Generalities on the Kerr geometry

In Boyer-Lindquist coordinates $x^\mu = (t, r, \theta, \phi)$, the Kerr metric is

$$ds^2 = -\frac{\Delta}{\Sigma}\left(dt - a\sin^2\theta d\phi\right)^2 + \Sigma\left(\frac{dr^2}{\Delta} + d\theta^2\right)$$
$$+ \frac{\sin^2\theta}{\Sigma}\left((r^2 + a^2)d\phi - a dt\right)^2 \tag{252}$$

with

$$\Delta(r) \triangleq r^2 - 2Mr + a^2, \quad \Sigma(r,\theta) \triangleq r^2 + a^2\cos^2\theta. \tag{253}$$

Assuming the validity of the cosmic censorship conjecture, the angular momentum per unit mass $a$ of the hole is bounded by its mass $M$: $|a| \leq M$. The Kerr metric admits a Killing-Yano tensor $Y_{\mu\nu} = Y_{[\mu\nu]}$ given by [43]

$$\frac{1}{2}Y_{\mu\nu}dx^\mu \wedge dx^\nu = a\cos\theta d\hat{r} \wedge (d\hat{t} - a\sin^2\theta d\hat{\phi})$$
$$+ \hat{r}\sin\theta d\theta \wedge \left((\hat{r}^2 + a^2)d\hat{\phi} - a d\hat{t}\right). \tag{254}$$

The derived Killing tensor $K_{\mu\nu} = K_{(\mu\nu)} = Y_\mu{}^\lambda Y_{\lambda\nu}$ obeys

$$K^{\mu\nu} = -2\Sigma\ell^{(\mu}n^{\nu)} - \hat{r}^2 g^{\mu\nu}, \tag{255}$$

where the two principal null directions of Kerr are given by $\ell^\mu\partial_\mu = \Delta^{-1}((\hat{r}^2 + a^2)\partial_{\hat{t}} + \Delta\partial_r + a\partial_{\hat{\phi}})$ and $n^\mu\partial_\mu = (2\Sigma)^{-1}((\hat{r}^2 + a^2)\partial_{\hat{t}} - \Delta\partial_{\hat{r}} + a\partial_{\hat{\phi}})$.

We introduce the tetrad[6] [44]

$$e^0_\mu \doteq \left(\sqrt{\tfrac{\Delta}{\Sigma}},\ 0,\ 0,\ -a\sin^2\theta\sqrt{\tfrac{\Delta}{\Sigma}}\right), \tag{256}$$

$$e^1_\mu \doteq \left(0,\ \sqrt{\tfrac{\Sigma}{\Delta}},\ 0,\ 0\right), \tag{257}$$

$$e^2_\mu \doteq \left(0,\ 0,\ \sqrt{\Sigma},\ 0\right), \tag{258}$$

$$e^3_\mu \doteq \left(-\tfrac{a}{\sqrt{\Sigma}}\sin\theta,\ 0,\ 0,\ \tfrac{r^2+a^2}{\sqrt{\Sigma}}\sin\theta\right). \tag{259}$$

We use lower-case Latin indices to denote tetrad components, e.g. $e^a_\mu = \left(e^0_\mu,\ e^1_\mu,\ e^2_\mu,\ e^3_\mu\right)$. The tetrad components $V^a$ of any vector **V** are given by $V^a = e^a_\mu V^\mu$. Tetrad indices are lowered and raised with the Minkowski metric $\eta_{ab} = \text{diag}(-1,\ 1,\ 1,\ 1)$. We choose the convention $\text{sign}(\varepsilon^{\hat{t}\hat{r}\theta\phi}) = +1$.

The Kerr spacetime admits two Killing vector fields, namely the time translation Killing field $k^\mu \doteq (1,\ 0,\ 0,\ 0)$ and the axis of axisymmetry, $l^\mu \doteq (0,\ 0,\ 0,\ 1)$.

In this tetrad basis, the KY tensor and its dual take the elegant form

$$\frac{1}{2}Y_{ab}dx^a \wedge dx^b$$
$$= a\cos\theta\, dx^1 \wedge dx^0 + r\, dx^2 \wedge dx^3, \tag{260}$$

$$\frac{1}{2}{}^*Y_{ab}dx^a \wedge dx^b$$
$$= r\, dx^1 \wedge dx^0 + a\cos\theta\, dx^3 \wedge dx^2. \tag{261}$$

### B.   Known quasi-conserved quantities

We now explicit the various quasi-invariants for the linearized MPT equations on Kerr spacetime. We recall that the norms of the vector-variables

$$\mu^2 = -p_a p^a, \qquad S^2 \triangleq S_a S^a \tag{262}$$

are *exactly* conserved along the motion.

*a.   Invariants generated by Killing fields.*   From the existence of the two Killing fields **k** and **l**, one can construct two linear invariants, namely the energy $E \triangleq -\mathcal{C}_k$ and the projection of the angular momentum along the direction of the BH spin, $\ell \triangleq \mathcal{C}_l$ where $\mathcal{C}_\xi$ is defined in Eq. (65). Their explicit expressions are given by [38]

---

[6] Following [27], we've introduced the symbol "$\doteq$", whose meaning is "the tensorial object of the left-hand side is represented in Boyer-Lindquist coordinates by the components given in the right-hand side". Similarly, we introduce the symbol "$\hat{=}$", bearing the same meaning, but in the tetrad basis.

$$E = E^G + \frac{M}{\Sigma^2}\left[\left(r^2 - a^2\cos^2\theta\right)S^{10} - 2ar\cos\theta S^{32}\right], \tag{263}$$

$$\ell = \ell^G + \frac{a\sin^2\theta}{\Sigma^2}\left[(r-M)\Sigma + 2Mr^2\right]S^{10}$$
$$+ \frac{a\sqrt{\Delta}\sin\theta\cos\theta}{\Sigma}S^{20} + \frac{r\sqrt{\Delta}\sin\theta}{\Sigma}S^{13} + \frac{\cos\theta}{\Sigma^2}\left[(r^2+a^2)^2 - a^2\Delta\sin^2\theta\right]S^{23}. \tag{264}$$

Here, $E^G$ and $\ell^G$ denote respectively the geodesic energy and angular momentum

$$E^G = -k_\mu p^\mu = -p_t = \sqrt{\frac{\Delta}{\Sigma}}p^0 + \frac{a\sin\theta}{\sqrt{\Sigma}}p^3, \tag{265}$$

$$\ell^G = l_\mu p^\mu = p_\phi = a\sin^2\theta\sqrt{\frac{\Delta}{\Sigma}}p^0 + (a^2+r^2)\frac{\sin\theta}{\sqrt{\Sigma}}p^3. \tag{266}$$

*b. Linear Rüdiger's quasi-invariant.* An elegant expression for $\mathcal{Q}_Y$ (66) is given through expressing the spin tensor components in the tetrad frame. Using Eq. (261), we find that Rüdiger's linear quasi-invariant is given by the simple expression

$$\mathcal{Q}_Y = 2\left(rS^{10} + a\cos\theta S^{32}\right). \tag{267}$$

*c. Quadratic Rüdiger's quasi-invariant.* Computing the various quadratic quasi-invariants that can be constructed amounts to evaluate the three scalar products $L_\alpha L^\alpha$, $S^\alpha \partial_\alpha \mathcal{Z}$ and $\xi_\alpha p^\alpha$. Three of those factors reduce to trivial expressions. First, because of the sign convention chosen for the relation between Killing and KY tensors, the product $L_\alpha L^\alpha$ reduces to the usual Carter constant $Q^G$,

$$L_\alpha L^\alpha = -K_{\alpha\beta}p^\alpha p^\beta = Q^G. \tag{268}$$

Second, a direct computation reveals that the vector $\xi^\alpha$ (138) reduces to the timelike Kerr Killing vector:

$$\xi^\alpha = k^\alpha \doteq (1,\ 0,\ 0,\ 0). \tag{269}$$

The product $\xi_\alpha p^\alpha$ is consequently equal to minus the geodesic energy $E^G$:

$$\xi_\alpha p^\alpha = -E^G. \tag{270}$$

Only the factor $S^\alpha \partial_\alpha \mathcal{Z}$ remains to be computed. The Killing-Yano scalar $\mathcal{Z}$ (170) reads

$$\mathcal{Z} = -ar\cos\theta. \tag{271}$$

The simplest expression for $S^\alpha \partial_\alpha \mathcal{Z}$ is provided through expressing the components of the spin vector in Boyer-Lindquist coordinates:

$$S^\alpha \partial_\alpha \mathcal{Z} = a(r\sin\theta\, S^\theta - \cos\theta\, S^r) \tag{272}$$

and it can be written explicitly as

$$S^\alpha \partial_\alpha \mathcal{Z} = -\frac{3a}{\mu\sqrt{\Sigma}}(\sqrt{\Delta}\cos\theta p^{[0}S^{23]} + r\sin\theta p^{[0}S^{13]}). \tag{273}$$

Rüdiger's quadratic invariant (225) consequently takes the form

$$\mathcal{Q}_R \triangleq -Q^G - 2\mu a\left(r\sin\theta S^\theta - \cos\theta S^r\right) - \mathcal{Q}_Y E^G. \tag{274}$$

In summary, we have in our possession four quasi-constants of the motion (in addition to $\mu^2$ and $\mathcal{S}$): $E$, $\ell$, $\mathcal{Q}_Y$ and $\mathcal{Q}_R$ that are conserved along the flow generated by the MPT equations at linear order in $S$. They consequently form a set of four linearly independent first integrals for the linearized MPT equations which, however, are *not* in involution, as will be later proven in Appendix A.

### C. Looking for mixed-symmetry tensors

Let us now address the question of the existence of a non-trivial mixed-symmetry Killing tensor, i.e. a tensor $T_{\alpha\beta\gamma}$ obeying the symmetries of a $\{2,1\}$ Young tableau and obeying Eq. (246), on Kerr spacetime. It is highly relevant because such the existence of such a tensor is in one-to-one correspondence with the existence of another independent quasi-conserved quantity that could, possibly, be in involution with the others first integrals. All the computations mentioned below being very cumbersome, they will not be reproduced here but are encoded in a *Mathematica* notebook, which is available on simple request.

From the symmetry in $(\beta\gamma)$ alone, there are $4 \times 10 = 40$ components in $T_{\alpha\beta\gamma}$ in 4 spacetime dimensions but they are not all independent because **T** is defined from **N** which is antisymmetric in its first indices. From the cyclic identity (245) we deduce the following: (i) The components of $T_{\alpha\beta\gamma}$ with all indices set equal to $i = 1,\ \ldots,\ 4$ is 0, $T_{iii} = 0$ (4 identities); (ii) In the presence of two distinct components $i, j = 1,\ \ldots,\ 4$ we have $2T_{iij} + T_{jii} = 0$ (12 identities); (iii) In the presence of three distinct components $i, j, k = 1,\ldots,4$ we have $T_{ijk} + T_{jki} + T_{kij} = 0$ (4 identities). There are no further algebraic symmetries. There are therefore 20 independent components, which we can canonically choose

to be the union of the sets of components $\mathcal{T}_2$ (of order $|\mathcal{T}_2| = 12$) and $\mathcal{T}_3$ (of order $|\mathcal{T}_3| = 8$) that are defined as

$$\mathcal{T}_2 = \{T_{ijj} | i, j = 1, \ldots, 4, \ j \neq i\}, \tag{275}$$
$$\mathcal{T}_3 = \{T_{123}, T_{213}, T_{124}, T_{214}, T_{134}, T_{314}, T_{234}, T_{324}\}. \tag{276}$$

The constraints (246) are 64 equations which can be splitted as follows: (i) 4 equations with 4 distinct indices $T_{i(jk;l)} = 0$; (ii) 12 equations with 3 distinct indices of type $T_{i(ij;k)} = 0$; (iii) 24 equations with 3 distinct indices of type $T_{i(jj;k)} = 0$; (iv) 12 equations with 2 distinct indices of type $T_{i(jj;j)} = 0$ and (v) 12 equations with 2 distinct indices of type $T_{i(ii;j)} = 0$.

Let us now specialize to the Kerr background and impose stationarity and axisymmetry, $\partial_t T_{\alpha\beta\gamma} = \partial_\phi T_{\alpha\beta\gamma} = 0$. In that case, the 4 equations $T_{r(tt;\phi)} = 0$, $T_{r(\phi\phi;t)} = 0$, $T_{\theta(tt;\phi)} = 0$, $T_{\theta(\phi\phi;t)} = 0$ and the 6 equations $T_{t(\phi\phi;\phi)} = 0$, $T_{r(\phi\phi;\phi)} = 0$, $T_{\theta(\phi\phi;\phi)} = 0$, $T_{r(tt;t)} = 0$, $T_{\theta(tt;t)} = 0$, $T_{\phi(tt;t)} = 0$ are algebraic, and can be algebraically solved for 10 out of the 20 variables. There are two additional combinations of the remaining equations that allow to algebraically solve for 2 further variables. After removing redundant equations, we can finally algebraically reduce the system of 64 equations in 20 variables to 13 partial differential equations in 8 variables.

Further specializing to the Schwarzschild background, we found the general solution to the thirteen equations. There are exactly two regular solutions which are both trivial mixed-symmetry tensors of the form (249) with either $\xi = \partial_t$ or $\xi = \partial_\phi$. Note that if we relax axisymmetry, there are two further trivial mixed-symmetry tensors (249) with $\xi$ given by the two additional $SO(3)$ vectors.

For Kerr, we did not find the general solution of the 13 partial differential equations in 8 variables. However, we obtained the most general perturbative deformation in $a$ of the 2-parameter family of trivial mixed-symmetry tensors of the form (249) with $\xi = \partial_t$ and $\partial_\phi$, assuming stationarity and axisymmetry. We obtained that the most general deformation is precisely a linear combination of the two trivial mixed-symmetry tensors of the form (249) with $\xi = \partial_t$ and $\partial_\phi$. Moreover, we also checked that there does not exist a consistent linear deformation of a linear combination of the two $\phi$-dependent $SO(3)$ Schwarzschild trivial mixed-symmetry tensors. Since we physically expect continuity of the quasi-conserved quantities between Kerr and Schwarzschild, we ruled out the existence of new stationary and axisymmetric quasi-conserved quantities of the MPT equations.

### D. Non-integrability of the linearized MPT system

We now stand in a comfortable position for discussing the integrability of the linearized MPT equations (55)-(56). Recall that an Hamiltonian system described by $N$ generalized coordinates $\{q^A\}$, their $N$ conjugated momenta $\{p_A\}$ and the Hamiltonian $H$ is said to be *completely integrable* (or integrable in the sense of Liouville) if there exist $N$ linearly independent first integrals of the motion $\{\mathcal{I}_A\}$ which are in involution, *i.e.* all the Poisson brackets between any two first integral should vanish [45]:

$$\{\mathcal{I}_A, \mathcal{I}_B\} = 0, \qquad \forall A, B \in 1, \ldots, N. \tag{277}$$

From an Hamiltonian perspective, the MPT equations form a system of dimension $N = 4$ [21]: in the momentum sector of the system, the constraint $\dot{\mu}^2 = 0$ leave us with three degrees of freedom. In the spin sector of the system, the constraints $\dot{\mathcal{S}} = 0$, $\mathcal{S}^* = 0$ and $p_\mu S^\mu = 0$ leave us with only one remaining degree of freedom. The evolution is driven by the Hamiltonian

$$H_{\text{MPT,lin}} = \frac{1}{2}g^{\mu\nu}p_\mu p_\nu = -\frac{\mu^2}{2} \tag{278}$$

together with the Poisson brackets [21, 46–51]

$$\{x^\mu, x^\nu\} = 0, \tag{279}$$
$$\{x^\mu, p_\nu\} = \delta^\mu_\nu, \tag{280}$$
$$\{p_\mu, p_\nu\} = -\frac{1}{2}R_{\mu\nu\kappa\lambda}S^{\kappa\lambda}, \tag{281}$$
$$\{S^{\mu\nu}, p_\kappa\} = 2\Gamma^{[\mu}_{\lambda\kappa}S^{\nu]\lambda}, \tag{282}$$
$$\{S^{\mu\nu}, x^\kappa\} = 0, \tag{283}$$
$$\left\{S^{\mu\nu}, S^{\kappa\lambda}\right\} = g^{\mu\kappa}S^{\nu\lambda} - g^{\mu\lambda}S^{\nu\kappa} + g^{\nu\lambda}S^{\mu\kappa} - g^{\nu\kappa}S^{\mu\lambda}. \tag{284}$$

Because it was shown that no additional independent quasi-conserved quantity could be found for the linearized MPT equations in Kerr, we can only exhibit the following set of four linearly independent non-trivial quasi-constants of motion:

$$\mathcal{I}_A = (E, \ell, \mathcal{Q}_Y, \mathcal{Q}_R). \tag{285}$$

The quantities $\mu^2$ and $\mathcal{S}$ are not included in this set since they are exactly conserved. The invariant mass is the Hamiltonian $H_{\text{MPT,lin}} = -\mu^2/2$ and the spin magnitude is a Casimir element of the Poisson bracket algebra, as shown in Appendix A.

The Poisson brackets between the integrals $\mathcal{I}_A$ are computed in Appendix A. After analysis, it is found that only the Poisson bracket $\{\mathcal{Q}_Y, \mathcal{Q}_R\}$ is non-vanishing at order $\mathcal{O}(\mathcal{S})$, as displayed in Table I. The four linearly independent first integrals $\mathcal{I}_A$ are consequently *not* in involution at the linear level, and the linearized MPT equations do not form an integrable system in the sense of Liouville.

### VIII. PERSPECTIVES

We reduced the existence of a new quasi-constant of motion of the Mathisson-Papapetrou-Tulczyjew equa-

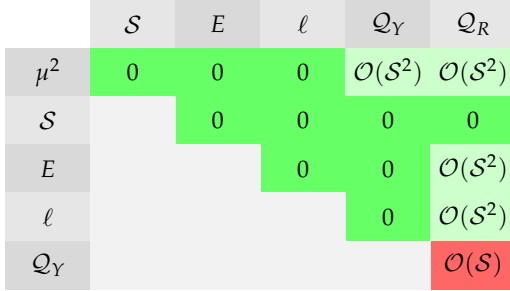

| | $\mathcal{S}$ | $E$ | $\ell$ | $\mathcal{Q}_Y$ | $\mathcal{Q}_R$ |
|---|---|---|---|---|---|
| $\mu^2$ | 0 | 0 | 0 | $\mathcal{O}(\mathcal{S}^2)$ | $\mathcal{O}(\mathcal{S}^2)$ |
| $\mathcal{S}$ | | 0 | 0 | 0 | 0 |
| $E$ | | | 0 | 0 | $\mathcal{O}(\mathcal{S}^2)$ |
| $\ell$ | | | | 0 | $\mathcal{O}(\mathcal{S}^2)$ |
| $\mathcal{Q}_Y$ | | | | | $\mathcal{O}(\mathcal{S})$ |

TABLE I. Poisson brackets between the first integrals $\mathcal{I}_A$ of linearized MPT equations. The bracket that is generally non-vanishing at order $\mathcal{O}(\mathcal{S})$ is represented in red, the other ones in green.

tions, *i.e.*, conserved at linear order in the spin, for Ricci-flat spacetimes admitting a Killing-Yano tensor to the existence of a mixed-symmetry Killing tensor on the background spacetime. The result applies in particular to Kerr spacetime, where it was shown that under the assumption of stationarity and axisymmetry only trivial mixed-symmetry Killing tensors exist which lead to products of known quasi-invariants. The absence of complete integrability in the sense of Liouville prevents us to rule out chaos at linear order in the spin, which is nevertheless suggested from the conclusions of [25, 27, 28]. An open question is how many of the two independent quasi-conserved quantities at linear order in the spin found by Rüdiger [22, 23] admit a generalization that is conserved at quadratic order. Even more interesting is whether a complete set of invariants can be built for the Mathisson-Papapetrou-Dixon equations including the relevant quadrupolar corrections to the MPT equations [18, 34, 35, 52, 53].

Even though we used the spin supplementary condition (SSC) of Tulczjew to derive our new constraint for the existence of additional non-trivial quasi-invariants, our reasoning also holds using the Mathisson-Pirani SSC [54, 55] or the Kyrian-Semerák SSC [56] as a direct consequence of Eq. (54). The explicit connection between the quasi-invariants detailed in this paper with the quasi-invariants of the spinning particle determined by a Grassmann odd spin [57, 58] remains to be established.

The occurrence of Einstein's tensor in the central identity (193) that we derived suggests that Rüdiger's quadratic invariant will admit a generalization to the Kerr-Newmann spacetime which also admits a Killing-Yano tensor, once the Einstein tensor is replaced with the electromagnetic stress-energy tensor. This remains to be investigated.

It also remains to be investigated whether the mixed-symmetry Killing tensors that we defined as trivial necessarily take the form of a sum of direct products of Killing(-Yano) vectors and tensors. Even if we suspect that this is true, we have not found a proof of such an assertion.

The Hamilton-Jacobi equation for the MPT equations was separated in [28] with separation constants of motion identical to the energy, angular momentum and the two Rüdiger quasi-invariants. Assuming no resonances, Witzany [28] was also able to derive action-angle variables and fundamental frequencies of the MPT system at linear order in the spin. However, the construction did not allow for a separation of the orbital equations of motion. This result is consistent with our findings that Liouville integrability does not hold due to a single non-vanishing Poisson bracket between the two Rüdiger quasi-invariants, even though the explicit relationship between these statements remains to be deepened.

## ACKNOWLEDGMENTS

We thank J. Vines and V. Witzany for pointing out two algebraic errors in a previous version of this manuscript which led to a major revision. We warmly thank L. Stein, J. Vines, V. Witzany for enlightening discussions during the Capra meeting 2021. A. D. warmly thanks his friends from Clerheid for having welcomed him during a large part of the period devoted to this work, and for having thus provided him with a wonderful working environment. A. D. is a Research Fellow and G. C. is Senior Research Associate of the F.R.S.-FNRS. G. C. acknowledges support from the FNRS research credit No. J003620F, the IISN convention No. 4.4503.15, and the COST Action GWverse No. CA16104.

## Appendix A: Poisson brackets

In this Appendix, we will compute all the independent Poisson brackets of the form $\{\mathcal{I}_A, \mathcal{I}_B\}$, with $\mathcal{I}_A = (E, \ell, \mathcal{Q}_Y, \mathcal{Q}_R)$. The computation below is build upon the fundamental relations (279)-(284). The results of the computations are summarized in Table I.

We begin by deriving an useful result about Poisson brackets: let $f(X)$ be an analytic function of some dynamical quantity $X$ such that the coefficients $f_n$ of the Taylor expansion $f(X) = \sum_{n=0}^{+\infty} f_n \frac{X^n}{n!}$ are constant, and let $Y$ be a dynamical quantity such that $\{X, Y\} \neq 0$. Then, from the Leibniz rule for Poisson brackets we have $\{X^n, Y\} = n X^{n-1}\{X, Y\}$ and one has the chain rule property

$$\{f(X), Y\} = \sum_{n=0}^{+\infty} \frac{f_n}{n!}\{X^n, Y\} \tag{A1}$$

$$= \sum_{n=1}^{+\infty} f_n \frac{X^{n-1}}{(n-1)!}\{X, Y\} \tag{A2}$$

$$= \frac{\mathrm{d}f}{\mathrm{d}X}\{X, Y\}. \tag{A3}$$

This relation is easily generalized to any analytic function of $n$ variables $X^\alpha$ ($\alpha = 1, \ldots, n$):

$$\{f(X^\alpha), Y\} = \frac{\partial f}{\partial X^\lambda}\{X^\lambda, Y\}. \tag{A4}$$

### 1. Semi-canonical basis

Before going further, notice that the Poisson brackets can be simplified through the impulsion's shift

$$p_\mu \to P_\mu \triangleq p_\mu + \chi_\mu, \qquad \text{with} \qquad \chi_\mu \triangleq \frac{1}{2}e_{va;\mu}e_b^v S^{ab}. \tag{A5}$$

The variables $x^\mu$ and $P_\nu$ are canonically conjugated[7] one to another. The only non-vanishing Poisson brackets are [21]

$$\{x^\mu, P_\nu\} = \delta_\nu^\mu, \tag{A6}$$

$$\{S^{ab}, S^{cd}\} = \eta^{ac}S^{bd} - \eta^{ad}S^{bc} + \eta^{bd}S^{ac} - \eta^{bc}S^{ad}. \tag{A7}$$

We can rewrite the second expression as

$$\{S^{ab}, S_{cd}\} = 4\delta_{[c}^{[a}S^{b]}{}_{d]}. \tag{A8}$$

### 2. $\{\mathcal{I}_A, \mathcal{S}\}$-type brackets

From the algebra (A6) and (A7), it is straightforward to notice that the quantities

$$\mathcal{S}^2 = \frac{1}{2}S^{ab}S_{ab}, \qquad (\mathcal{S}^*)^2 = \varepsilon_{abcd}S^{ab}S^{cd} \tag{A9}$$

are Casimir element of the algebra, *i.e.* their Poisson brackets with all other dynamical variables are vanishing, as noticed in [21]. Under the Tulczyjew spin supplementary condition (13), one has $\mathcal{S}^* = 0$ as a direct consequence of (22) and we shall not consider $\mathcal{S}^*$ any further. The property (A9) together with the chain rule

$$\{\mathcal{I}_A, \mathcal{S}\} = \frac{1}{2\mathcal{S}}\{\mathcal{I}_A, \mathcal{S}^2\} \tag{A10}$$

leads to

$$\{\mathcal{I}_A, \mathcal{S}\} = 0. \tag{A11}$$

### 3. $\{\mathcal{I}_{\hat{A}}, E\}$-type brackets

*a.* $\mathcal{I}_{\hat{A}} = \ell$. Using the identities $\nabla_\alpha k^\mu = \Gamma_{\alpha t}^\mu$ and $\nabla_\alpha l^\mu = \Gamma_{\alpha\phi}^\mu$, the bracket reads

$$\{\ell, E\} = \{p_t, p_\phi\} + \frac{1}{2}\nabla_\alpha k_\beta\{S^{\alpha\beta}, p_\phi\} - \frac{1}{2}\nabla_\alpha l_\beta\{S^{\alpha\beta}, p_t\} - \frac{1}{4}\nabla_\alpha l_\beta\nabla_\gamma k_\delta\{S^{\alpha\beta}, S^{\gamma\delta}\} \tag{A12}$$

$$= -\frac{1}{2}R_{t\phi\alpha\beta}S^{\alpha\beta} + \left(\nabla_\alpha l_\beta \Gamma_{\lambda t}^\alpha - \nabla_\alpha k_\beta \Gamma_{\lambda\phi}^\alpha\right)S^{\lambda\beta} + \nabla_\alpha l^\lambda \nabla_\lambda k_\beta S^{\alpha\beta} \tag{A13}$$

$$= -\frac{1}{2}R_{t\phi\alpha\beta}S^{\alpha\beta} - \nabla_\alpha l^\lambda \nabla_\lambda k_\beta S^{\alpha\beta} = 0. \tag{A14}$$

The last equality follows from the fact that the axisymmetry of Kerr spacetime together with the definition of the Riemann tensor enforce the relation

$$R_{t\phi\alpha\beta}S^{\alpha\beta} = 2\Gamma_{t\lambda}^\alpha\Gamma_{\phi\beta}^\lambda S_\alpha{}^\beta = -2\nabla_\alpha l^\lambda\nabla_\lambda k_\beta S^{\alpha\beta} \tag{A15}$$

to hold.

*b.* $\mathcal{I}_{\hat{A}} = \mathcal{Q}_Y$. The axisymmetric character of Kerr spacetime enforces the tetrad $e_a^\mu$ to be independent of $t$ and $\phi$. This yields $\{p_{t,\phi}, e_a^\mu\} = -\partial_{t,\phi}e_a^\mu = 0$. Conse-

quently,

$$\{p_{t,\phi}, S^{ab}\} = \{p_{t,\phi}, S^{\mu\nu}\}e_\mu^a e_\nu^b. \tag{A16}$$

Using this last equation and the fact that $\mathcal{Q}_Y$ commute with $x^\mu$, we get

$$\{\mathcal{Q}_Y, E\} = \{p_t, \mathcal{Q}_Y\} + \frac{1}{2}\nabla_a k_b\{S^{ab}, \mathcal{Q}_Y\} \tag{A17}$$

$$\overset{(A16)}{=} -4\left[r\left(\Gamma_{kt}^{[1} - \nabla_k k^{[1}\right)S^{0]k}\right.$$

$$\left. + a\cos\theta\left(\Gamma_{kt}^{[3} - \nabla_k k^{[3}\right)S^{2]k}\right] \tag{A18}$$

$$= 0. \tag{A19}$$

---

[7] It is also possible to introduce canonical coordinates for the spin sector of the system, see e.g. [28] for details.

*c.* $\mathcal{I}_{\hat{A}} = \mathcal{Q}_R$. Using the identity

$$\left\{E^G, E\right\} = \frac{1}{2}\nabla_\alpha k_\beta \Gamma^\alpha_{\lambda t} S^{\lambda\beta} \tag{A20}$$

$$= -\frac{1}{2}\nabla_\beta k_\alpha \nabla_\lambda k^\alpha S^{\lambda\beta} = 0 \tag{A21}$$

and the chain rule (A4), one has

$$\{\mathcal{Q}_R, E\} = 2p_\mu K^{\mu\nu}\{p_\nu, E\} - 2\mu\{S^\alpha \partial_\alpha \mathcal{Z}, E\} \tag{A22}$$

$$= -p_\mu K^{\mu\nu}\left(2\{p_\nu, p_t\} + \left\{p_\nu, \nabla_\alpha k_\beta S^{\alpha\beta}\right\}\right)$$
$$+ \mu\left(2\{S^\alpha \partial_\alpha \mathcal{Z}, p_t\} + \{S^\alpha \partial_\alpha \mathcal{Z}, \nabla_\mu k_\nu S^{\mu\nu}\}\right). \tag{A23}$$

Making use of the identity

$$R_{t\beta\mu\nu} S^{\mu\nu} = \partial_\beta\left(\nabla_\mu k_\nu\right)S^{\mu\nu} + 2\nabla_\alpha k_\nu \Gamma^\alpha_{\beta\lambda} S^{\nu\lambda}, \tag{A24}$$

the two first Poisson brackets of this expression can be shown to cancel mutually, and we are left with

$$\{\mathcal{Q}_R, E\} = \frac{1}{2}\varepsilon^{\alpha\beta\gamma\delta}S_{\gamma\delta}\,\partial_\alpha \mathcal{Z}\left[R_{t\beta\mu\nu} - \partial_\beta\left(\nabla_\mu k_\nu\right)\right]S^{\mu\nu} \tag{A25}$$

$$= \varepsilon^{\alpha\beta\gamma\delta}\partial_\alpha \mathcal{Z}S_{\gamma\delta}\nabla_\rho k_\nu \Gamma^\rho_{\lambda\beta} S^{\nu\lambda} \tag{A26}$$

$$= \mathcal{O}(\mathcal{S}^2). \tag{A27}$$

#### 4. $\{\mathcal{I}_A, \ell\}$-type brackets

The computations are identical to the $\{\mathcal{I}_A, E\}$-type case, but with $k^\alpha \to l^\alpha$. We consequently find

$$\{\mathcal{Q}_Y, \ell\} = 0, \tag{A28}$$
$$\{\mathcal{Q}_R, \ell\} = \mathcal{O}(\mathcal{S}^2). \tag{A29}$$

#### 5. $\{\mathcal{I}_A, \mathcal{Q}_Y\}$-type brackets

The final bracket to be computed takes the form

$$\{\mathcal{Q}_R, \mathcal{Q}_Y\} = 2p_\mu K^{\mu\nu}\{p_\nu, \mathcal{Q}_Y\} - 2\mu\,\partial_\alpha \mathcal{Z}\{S^\alpha, \mathcal{Q}_Y\} \tag{A30}$$

$$= -2p_\mu K^{\mu\nu}\left(\partial_\nu Y^*_{\alpha\beta}S^{\alpha\beta} + 2\Gamma^\alpha_{\rho\nu}S^{\beta\rho}Y^*_{\alpha\beta}\right) - 4\varepsilon^{\alpha\beta}{}_{\gamma\delta}\partial_\alpha \mathcal{Z}p_\beta Y^{*\gamma}{}_\nu S^{\delta\nu} + \mathcal{O}(\mathcal{S}^2) \tag{A31}$$

$$= -2p_\mu K^{\mu\nu}\nabla_\nu Y^*_{\alpha\beta}S^{\alpha\beta} - 2\varepsilon^{\alpha\beta\gamma\delta}\partial_\alpha \mathcal{Z}p_\beta\varepsilon_{\gamma\nu\rho\sigma}Y^{\rho\sigma}S_\delta{}^\nu + \mathcal{O}(\mathcal{S}^2) \tag{A32}$$

$$= 4p_\mu S^{\alpha\beta}\left(K^\mu{}_\alpha\xi_\beta + Y^\mu{}_\alpha Y^\lambda{}_\beta\xi_\lambda\right) + \mathcal{O}(\mathcal{S}^2). \tag{A33}$$

This expression is generally non-vanishing at order $\mathcal{O}(\mathcal{S})$.

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
