# Peer review of "Complete set of quasi-conserved quantities for spinning particles around Kerr"

_SciPost Physics_

## Round 2 · Referee Report · Anonymous (Referee 1) · 2021-8-15

Strengths

  1. The article addresses a very important question.
  2. The article performs a strong technical analysis.
  3. The article is very well written.

Weaknesses

The paper has no weaknesses.

Report

This very interesting article studies conserved quantities for spinning particles in spaces admitting Killing-Yano tensors. Then the authors apply their general formalism to a particular case of the Kerr black hole, which gives the most interesting physical example of geometries with Killing-Yano tensors. The authors derive the restrictions on the spacetime that guarantee Liouville integrability of the equations of motion for spins, and they demonstrate that such restrictions are satisfied by the Kerr geometry. The article addresses a very important question in general relativity, and it is very well written, so I recommend it for publication.

I would recommend one addition that would be beneficial to the readers. The article reduces the criterion for integrability to existence of "mixed-symmetry Killing tensors", and they demonstrate that such objects don't exist for the Kerr geometry. It would be interesting to add a small section before VII with some examples of spaces with mixed-symmetry tensors and the resulting integrability. The examples can be as simple as flat space, but I think that they would provide an excellent illustration of the formalism. I leave this addition to the authors' discretion, and I recommend this paper for publication.

Requested changes

The proposed optional change is mentioned in the report.

  • validity: top
  • significance: high
  • originality: high
  • clarity: top
  • formatting: perfect
  • grammar: perfect

Author:  Adrien Druart  on 2021-09-06  [id 1739]

(in reply to Report 1 on 2021-08-15)

We thank the referee for his/her comments and suggestion. We plan to improve the manuscript with the proof that all mixed-symmetry Killing tensors of Minkowski spacetime are trivial. We will leave the investigation of mixed-symmetry Killing tensors for (anti-)de Sitter spacetimes for further work since it is not really related to the topic of this paper.

---

## Round 2 · Referee Report · Anonymous (Referee 2) · 2021-9-1

Report

Authors ask on existence of linear on basic variables conserved quantities for Dixon spinning body in Kerr space, in addition to the energy and angular momentum. From general considerations, I am rather skeptical on existence of such quantities, so the kind of "negative report", presented by authors, is not surprising. Nevertheless, their analysis of Killing-Yano tensors is of interest, and the work can be published in SciPost Physics in the present form.

Requested changes

  1. Author's explanation for their notion of quasi-conserved quantity on page 1 is rather confusing: "They are quasi-conserved in the sense that they are no longer conserved at quadratic order in the spin without further corrections". I encourage authors to give the exact and clear definition of what they mean by "quasi-conserved" quantity.

  • validity: -
  • significance: -
  • originality: -
  • clarity: -
  • formatting: -
  • grammar: -

Author:  Adrien Druart  on 2021-09-06  [id 1738]

(in reply to Report 2 on 2021-09-01)

We thank the referee for his/her comment. We plan to improve the manuscript with a rephrased the definition of quasi-conservation.

---

## Round 3 · Author Response

We took into account the comments of both referees. The manuscript was improved with the minor changes listed below.

---

## Round 3 · List of Changes

- A rephrased the definition of quasi-conservation;
- The proof that all mixed-symmetry Killing tensors of Minkowski spacetime are trivial. We leave the investigation of mixed-symmetry Killing tensors for (anti-)de Sitter spacetimes for further work since it is not really related to the topic of this paper.

---

## Editorial Decision

publication_decision_taken:_accept